

# Impulse response functions as a framework for quantifying ocean-based carbon dioxide removal

Elizabeth Yankovsky[1,4], Mengyang Zhou[2], Michael Tyka[3], Scott Bachman[1,6], David T. Ho[1,5], Alicia Karspeck[1], and Matthew C. Long[1,6]

[1][C]Worthy, LLC, Boulder, CO 80302, USA
[2]University of Connecticut, Groton, CT 06340, USA
[3]Google Inc., Seattle, WA 98103, USA
[4]Yale University, New Haven, CT 06511, USA
[5]University of Hawai'i at Mānoa, Honolulu, HI 96822, USA
[6]NSF National Center for Atmospheric Research, Boulder, CO 80307, USA

**Correspondence:** Elizabeth Yankovsky (elizabeth.yankovsky@yale.edu)

**Abstract.** Limiting global warming to 2°C by the end of the century requires dramatically reducing $CO_2$ emissions, and also implementing carbon dioxide removal (CDR) technologies. A promising avenue is marine CDR through ocean alkalinity enhancement (OAE). However, quantifying carbon removal achieved by OAE deployments is challenging because it requires determining air-to-sea $CO_2$ transfer over large spatiotemporal scales—and there is the possibility that ocean circulation will

remove alkalinity from the surface ocean before complete equilibration. This challenge makes it difficult to establish robust accounting frameworks suitable for an effective carbon market. Here, we propose using impulse response functions (IRFs) to address such challenges. We perform model simulations of a short-duration alkalinity release (the "impulse"), compute the resultant air-sea $CO_2$ flux as a function of time, and generate a characteristic carbon uptake curve for the given location (the IRF). Applying the IRF method requires a linear and time-invariant system. We attempt to meet these conditions by using

small alkalinity forcing values and creating an IRF ensemble accounting for seasonal variability. The IRF ensemble is then used to predict carbon uptake for an arbitrary-duration alkalinity release at the same location. We test whether the IRF approach provides a reasonable approximation by performing OAE simulations in a global ocean model at locations that span a variety of dynamical and biogeochemical regimes. We find that the IRF prediction can typically reconstruct the carbon uptake in continuous-release simulations within several percent error. Our simulations elucidate the influences of oceanic variability and

deployment duration on carbon uptake efficiency. We discuss the strengths and possible shortcomings of the IRF approach as a basis for quantification and uncertainty assessment of OAE, facilitating its potential for adoption as a component of the carbon removal market's standard approach to Monitoring, Reporting, and Verification (MRV).

## 1 Introduction

Limiting global warming to 1.5 or 2°C as outlined in the Paris Agreement necessitates the rapid reduction in $CO_2$ emissions in

conjunction with the deployment of carbon dioxide removal (CDR) technologies (Rogelj et al., 2018). The scenarios considered in the Intergovernmental Panel on Climate Change report aligned with such climate goals require a total amount of CDR on the





order of 100-1000 Gt of $CO_2$ through the end of this century (IPCC, 2023). An increasing amount of research aims to develop a portfolio of terrestrial and marine CDR (mCDR) methods and explore their viability, scalability, and complex influences on the Earth system (Nemet et al., 2018; Shepherd, 2009). Earth's atmosphere, hydrosphere, biosphere, and lithosphere are all
open systems that store and exchange carbon with one another. Evaluating mCDR effects thus necessitates a broad process-level understanding of all constituents of the carbon cycle. However, there remain critical gaps in both (1) the current state of knowledge regarding mCDR influences within the global carbon cycle and (2) the theoretical, observational, and modeling tools underpinning a tractable mCDR assessment framework.

Ocean-based CDR is hypothesized to have climatically relevant scalable potential and ocean alkalinity enhancement (OAE)
offers an advantageous mCDR approach as it attempts to accelerate a natural process on Earth (Gernon et al., 2021). A "thermostat" operates on 1-10 million year timescales created by the weathering of carbonate and silicate minerals on land and the subsequent deposition of carbonate minerals in the ocean. When atmospheric $CO_2$ is high, surface temperature increases and the hydrological cycle strengthens, leading to increased chemical weathering. This weathering increases the alkalinity of the surface ocean and draws $CO_2$ out of the atmosphere, eventually lowering surface temperatures and leading to a phase reversal
in the feedback (Walker et al., 1981; Berner et al., 1983). Natural weathering sequesters on the order of 1 Gt of $CO_2$ per year (Gaillardet et al., 1999). By artificially supplying alkalinity via OAE, a deficit in the partial pressure of $CO_2$ ($pCO_2$) is generated in the surface ocean. Gas exchange then leads to a flux of $CO_2$ from the atmosphere into the ocean, thus speeding up the natural uptake of atmospheric $CO_2$. However, ocean circulation and turbulent mixing lead to lateral and vertical transport of alkalinity. If alkalinity is subducted away from the surface, the $pCO_2$ deficit will not lead to additional $CO_2$ uptake
by the ocean. Ocean dynamics are thus crucial to modulating the efficiency of mCDR interventions and pose a significant quantification challenge for OAE.

Developing simplified models of the factors governing OAE and employing statistical techniques to reduce the problem's complexity is a major objective in facilitating successful Monitoring, Reporting, and Verification (MRV) of CDR interventions. MRV refers to the development of standards and practices evaluating CDR influences in a systematic, unbiased way across
sectors (Reinhard et al., 2023; Fuss et al., 2014). MRV of interventions such as afforestation, macroalgae cultivation, artificial ocean upwelling, ocean iron fertilization, OAE, and solar radiation management has been attempted by studies such as Keller et al. (2014), but it remains clear that CDR quantification is plagued by uncertainty. Furthermore, although land-based CDR has relatively established MRV practices (Brack and King, 2021), marine-based approaches are still in their infancy (Oschlies et al., 2023; Ho et al., 2023). Developing MRV for mCDR is particularly urgent as mCDR commences its rapid emergence into
the carbon market (Bach et al., 2023). MRV for mCDR is challenging due to the multiscale nature of ocean dynamics spanning molecular to global and nanosecond to millennial scales (Renforth and Henderson, 2017). For example, an OAE deployment may occur on the scale of meters but will be transported by the flow, experience gas exchange and potential feedbacks mediated by biota as it disperses, and will spread to basin and global scales over years/decades. A single MRV strategy is thus insufficient to capture all of the complex spatiotemporal influences of an mCDR intervention. From an observational perspective, mCDR is
characterized by unfavorable signal-to-noise ratios extending over vast spatial scales. The infeasibility of complete observation-based MRV necessitates numerical modeling, which is also problematic due to the inherent difficulties in representing the ocean





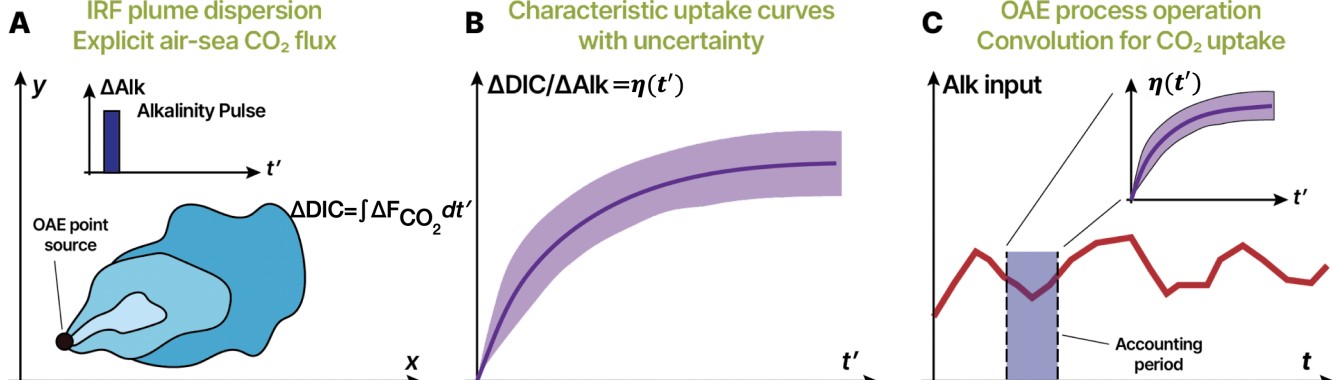

**Figure 1.** Overview of the IRF approach. A: An alkalinity perturbation is applied as a pulse to the surface ocean from a point source; an alkalinity plume spreads. There is an increase in DIC given by the integrated flux of carbon dioxide into the ocean from the OAE intervention ($\Delta F_{CO_2}$). B: The carbon uptake yields a characteristic uptake curve $\eta$. C: For an arbitrary OAE deployment, a convolution of the forcing time series and characteristic uptake curve may be performed to estimate the carbon uptake for accounting purposes.

component of the climate and its interactions with the atmosphere, land, and biosphere using finite computational resources (Ho et al., 2023). Furthermore, baselines, or counterfactual experiments are needed to assess the influence of mCDR interventions.

Here, we develop the idea of using impulse response functions (IRFs) as a statistical MRV tool for mCDR. We use OAE as a testbed for the IRF approach, but note this methodology should be suitable for other mCDR intervention strategies, such as direct ocean removal. A useful metric for quantifying OAE efficiency as a function of time, $\eta(t)$ is:

$$\eta(t) = \frac{\Delta \text{DIC}(t)}{\Delta \text{Alk}(t)}, \tag{1}$$

where $\Delta \text{Alk}(t)$ is the net amount of excess alkalinity put into the system through the OAE intervention at a given time, and $\Delta \text{DIC}(t)$ is the increase of ocean dissolved inorganic carbon (DIC) due to the intervention as a function of time. There is a theoretical maximum $\eta$ that varies between about 0.7 and 0.9 over the global ocean, which depends on temperature, salinity, and carbonate chemistry (Sarmiento and Gruber, 2006). Ocean dynamics, such as convection or vertical mixing, can alter the actual efficiency value from the theoretical value obtained solely from thermodynamics.

Figure 1 provides an overview of the proposed IRF approach. In Figure 1, an "impulse" of alkalinity forcing is applied as a pulse of a given $\Delta \text{Alk}$ at the surface from a point source. The alkalinity plume spreads spatially over time away from the point source. The OAE release has an associated characteristic uptake curve – $\eta(t)$, which we call the "impulse response function" – with an envelope of uncertainty due to variability of the system that might be characterized by repeating the IRF experiment multiple times to sample differing winds, seasonal variability, mixing dynamics, biological activity, etc. If the system is sufficiently linear and time-invariant (detailed in the next section), we may perform a convolution of this uptake curve with any arbitrary alkalinity forcing to obtain the resultant uptake curve for that forcing. Figure 1C illustrates a continuous OAE deployment as a function of time. Computing carbon uptake owing to alkalinity released during the accounting period





highlighted in purple would involve performing a convolution with alkalinity forcing over that time period and the characteristic curve shown in Figure 1B.

## 2 Background on Impulse Response Functions

The validity of IRF-based CDR quantification is based upon two conditions: linearity and time invariance. Consider a system
governed by a particular equation or set of equations, such as the ocean; an input, $x$, is supplied to the system, resulting in an
output $y$. For mathematical purposes, $x$ and $y$ are arbitrary, but we can imagine $x$ in our problem to be an alkalinity forcing
$x(t) = \Delta\mathrm{Alk}(t)$ and $y$ to be the resultant carbon uptake efficiency curve $y(t) = \eta(t)$. If we obtain an output $y_1$ for input $x_1$,
and $y_2$ for input $x_2$, then the linearity condition states that for an input $x_1 + x_2$ we will obtain an output $y_1 + y_2$. A linear system
should also obey homogeneity, such that for input $ax_1 + bx_2$ we should obtain output $ay_1 + by_2$. The second condition is time
invariance. If we perturb our system with an impulse-like input (in practice, this will be applied over a discrete but small time
interval) at different points in time and obtain the corresponding impulse response functions, or outputs, $h_1(t)$, $h_2(t)$, $h_3(t)$...,
these IRFs should be identical: $h_1(t) = h_2(t) = h_3(t)$.... In other words, the IRF is stationary in time. A system satisfying both
of these conditions is known as linear and time invariant (LTI).

We make use of the Dirac delta function $\delta(t)$, which is a function that is zero everywhere except at the origin $t = 0$, where
its value is one:

$$\delta(t) = \begin{cases} 1, & t = 0 \\ 0, & t \neq 0 \end{cases}.$$

The idea of a convolution is to take an arbitrary signal $x(t)$ and express it as a sum of simpler signals, i.e., a weighted sum of
impulses. At $t = 0$, we express $x(t)$ as $x(0)$ multiplied by the Dirac delta function, and likewise for all subsequent values of $t$
(with appropriately shifted delta functions) so that:

$$x(t) = \sum_{t'=-\infty}^{\infty} x(t')\delta(t - t'). \tag{2}$$

Now, let us consider the output $y(t)$ to an input $x(t)$ of an LTI system. The IRF, denoted as $h(t)$, is a known function describing
how our system responds to an impulse. Extending the convolution idea, we rewrite our output $y(t)$ as a weighted sum of
impulse responses, so that:

$$y(t) = \sum_{t'=-\infty}^{\infty} x(t')h(t - t') = \int_{-\infty}^{\infty} x(t')h(t - t')dt' = x(t) * h(t). \tag{3}$$

The integral form is presented on the right-hand side. We have rewritten a complex function $y(t)$ in terms of weights $x(t)$ and
impulse responses $h(t)$. Note that linearity allows us to sum/integrate over all the individual inputs, and time invariance allows
us to conclude that if the response to $\delta(t)$ is $h(t)$, then the response to $\delta(t - 1)$ is $h(t - 1)$. For the OAE problem, we consider





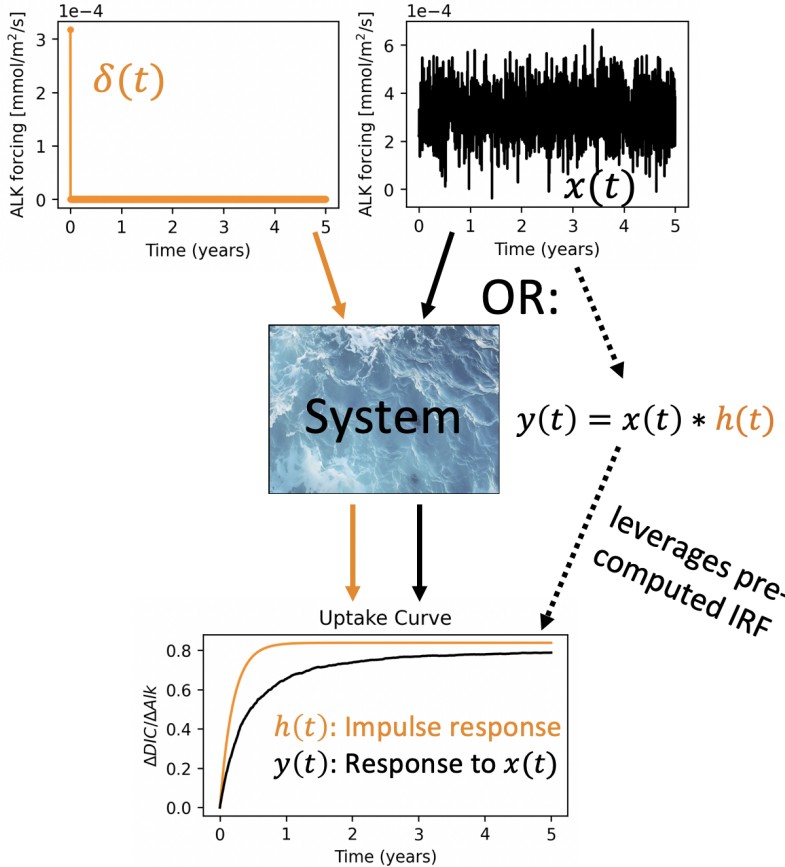

**Figure 2.** An illustration of why IRFs provide a powerful statistical and data reduction tool for the OAE problem. We consider our system to be the combination of chemical, biological, and physical processes influencing carbon uptake and distribution in the ocean. We probe the system by performing a model integration with a pulse $\delta(t)$ of alkalinity forcing and computing the resultant impulse response function $h(t)$ defined as $\eta(t)$ (OAE efficiency). Next, we want to obtain the OAE efficiency curve $y(t)$ for a continuous five-year alkalinity deployment $x(t)$. One option is to perform the full model integration for this OAE deployment and compute the resultant OAE efficiency. However, provided we have a sufficiently LTI system, we can compute the convolution of the IRF and the forcing, thus avoiding the need for an additional model integration.

$x$ to be the alkalinity forcing and $y$ to be the resulting carbon uptake efficiency, so that Equation 3 becomes:

$$\eta(t) = \int_{-\infty}^{\infty} \Delta\mathrm{Alk}(t')h(t-t')dt' = \Delta\mathrm{Alk}(t) * h(t). \tag{4}$$

One may ask: What is the advantage of rewriting $y(t)$ or $\eta(t)$ in the form of Equations 3-4? We address this question in Figure 2. The IRF approach allows us to obtain the system's response $y(t)$ for **any** arbitrary input $x(t)$, provided that we know the IRF and that the ocean is acceptably well-approximated as an LTI system. In other words, if we obtain the characteristic



uptake curve(s) for an impulse of alkalinity forcing, we can then predict the uptake curve $\eta(t)$ for any time series of alkalinity forcing $\Delta\mathrm{Alk}(t)$. This is powerful because it implies that we no longer have to explicitly simulate the fate of every mole of

alkalinity that we put into the ocean. We integrate the model to compute the IRF and then perform convolutions to predict the uptake for any alkalinity input. Of course, this simplification relies on staying within the bounds of LTI. Although we know that the ocean is not LTI, a major question that we address in this work is whether the statistics of the system are sufficiently stationary to allow us to apply the IRF approach via either a single IRF or library of IRFs encapsulating seasonal variability.

    Several studies in the climate science field have successfully applied the IRF methodology to gain insight into various sys-

tems. Kuang (2010) used linear response functions to study deep cumulus cloud convection, which involves many nonlinear processes. Nonetheless, an ensemble can be used to create a reference state, and then linear response functions to small perturbations around this mean state were used to probe the dynamics of the cumulus ensemble and develop a parameterization. Joos et al. (2013) used IRFs to characterize the responses of metrics such as Global Warming Potential and Global Temperature Change Potential to emission pulses of $CO_2$ into the atmosphere. Hassanzadeh and Kuang (2016) used Green's functions (note

that IRFs are a special kind of Green's functions with zero initial conditions) to determine the mean response of the climate system to weak forcing. They applied the response function alongside an eddy flux matrix to study eddy-mean flow interactions and gain insights into complex eddy feedbacks. All of these studies identify clear bounds under which the LTI assumption holds and the IRF approach is valid – doing so for the OAE problem is the subject of the next section.

## 3   Applying IRFs to the OAE Problem

The viability of IRFs in developing quantitative estimates of CDR from OAE relies on satisfying two conditions: linearity and time invariance. We consider chemical, physical, and biological controls on OAE uptake efficiency. We begin by viewing the OAE problem as a purely chemical process neglecting oceanic dynamics, i.e., any processes that may advect, disperse, or subduct added alkalinity and thus modify the associated OAE efficiency curve, as well as biological feedbacks onto the carbonate system.

### 3.1   Chemistry

From a chemical perspective, OAE operates by inducing a $p\mathrm{CO}_2$ deficit at the surface ocean that is subsequently relaxed at a rate proportional to the gas exchange kinetics. The OAE efficiency $\eta(t)$ may be expressed as a simple exponential (Zeebe and Wolf-Gladrow, 2001) as:

$$\eta(t) = \eta_{max}e^{-t/\tau}. \tag{5}$$

There are two parameters controlling the uptake efficiency. $\eta_{max}$ is typically 0.7-0.9 depending on the background carbonate system. $\tau$ is the timescale of equilibration determined by gas exchange (Sarmiento and Gruber, 2006; Jones et al., 2014; He and Tyka, 2023) defined as:

$$\tau = \frac{\partial\mathrm{DIC}}{\partial p\mathrm{CO}_2} \cdot \left(\frac{h}{K_0 k_w}\right), \tag{6}$$



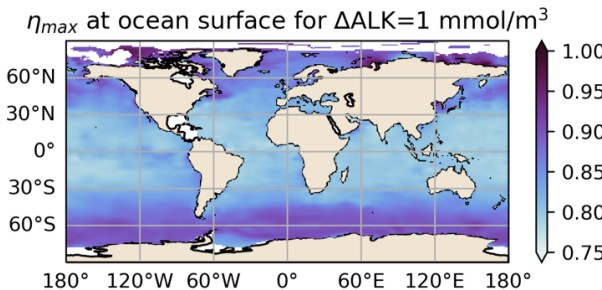 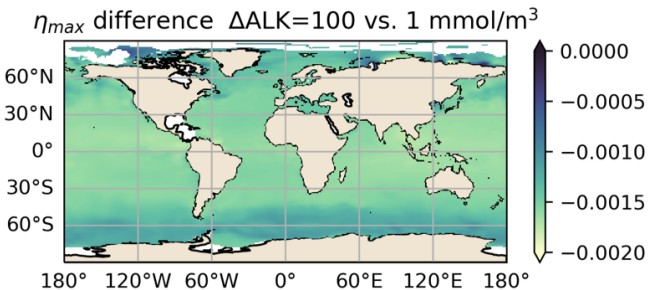

**Figure 3.** Left: values of the maximum theoretical OAE efficiency $\eta_{max}$ for the global ocean based on GLODAPv2 observations (Lauvset et al., 2016; Olsen et al., 2016) for a $\Delta$Alk perturbation of 1 mmol/m$^3$. Right: the difference in $\eta_{max}$ for $\Delta$Alk of 100 mmol/m$^3$ and 1 mmol/m$^3$.

where $K_0$ is the CO$_2$ solubility coefficient (Weiss, 1974), $h$ is mixed layer depth, $k_w$ is the gas transfer velocity (Ho et al., 2006;
Wanninkhof, 2014), and $p$CO$_2$ is the partial pressure of CO$_2$ (Sarmiento and Gruber, 2006). The linearity assumption states
that the uptake curve $\eta(t)$ should obey superposition – if two alkalinity pulses are applied independently, then $\eta(t)$ should be
the same as the case where the alkalinity pulses are applied together. This also means that both $\eta_{max}$ and $\tau$ cannot be functions
of $\Delta$Alk. Otherwise, $\eta(t)$ in Equation 5 would nonlinearly depend on $\Delta$Alk. We know this is not the case, but we aim to
quantify the expected error and assess whether it is sufficiently small. We employ observed carbonate variables synthesized
by the Global Ocean Data Analysis Project (GLODAPv2) (Lauvset et al., 2016; Olsen et al., 2016), to sample the full range
of carbonate system conditions across the global surface ocean. We take the mean Alk and DIC state at every grid point and
then apply perturbations to alkalinity (i.e., $\Delta$Alk) ranging from 1 to 100 with intervals of 10 mmol/m$^3$. For reference, the
forcing employed in this study is $3.17 \cdot 10^{-4}$ mmol/m$^2$/day (or 10 mol/m$^2$/year). We then compute the corresponding $\eta_{max}$ and
$\tau$ as functions of temperature, salinity, phosphate, silicate, perturbed Alk, and DIC values based on the governing equations of
the carbonate system (Sarmiento and Gruber, 2006) and ask whether they change substantially as the alkalinity perturbation
increases.

We first test the linearity condition for $\eta_{max}$ in Figure 3. We note that the maximum theoretical OAE efficiency varies from
0.75 at low latitudes to 0.9 at high latitudes. Looking at the difference between the $\Delta$Alk = 100 and 1 mmol/m$^3$ cases, there
appears to be only a slight decrease in $\eta_{max}$ as the forcing increases. To compare more quantitatively, we create a parameter
space of DIC and Alk based on surface climatology and compute the $\eta_{max}$ for each value of the $\Delta$Alk perturbation. We
compute the slope of the resulting curve to obtain the change in $\eta_{max}$ per mmol/m$^3$ of added Alk. The result is plotted in the
left panel of Figure 4. The percent change in $\eta_{max}$ for 100 mmol/m$^3$ of added Alk is less than 0.1%, nearly negligible for the
entire parameter space considered here. Given the expected magnitudes of Alk perturbations induced by CDR deployments, it
is unlikely that we will introduce alkalinity perturbations in excess of 100 mmol/m$^3$ at the CO$_2$ equilibration scales following
dilution and spread of the alkalinity plume (though possible in the vicinity of the OAE deployment). $\eta_{max}$ is essentially
unaffected for such a perturbation. Thus, we conclude that $\eta_{max}$ does not present significant nonlinearity. We then consider





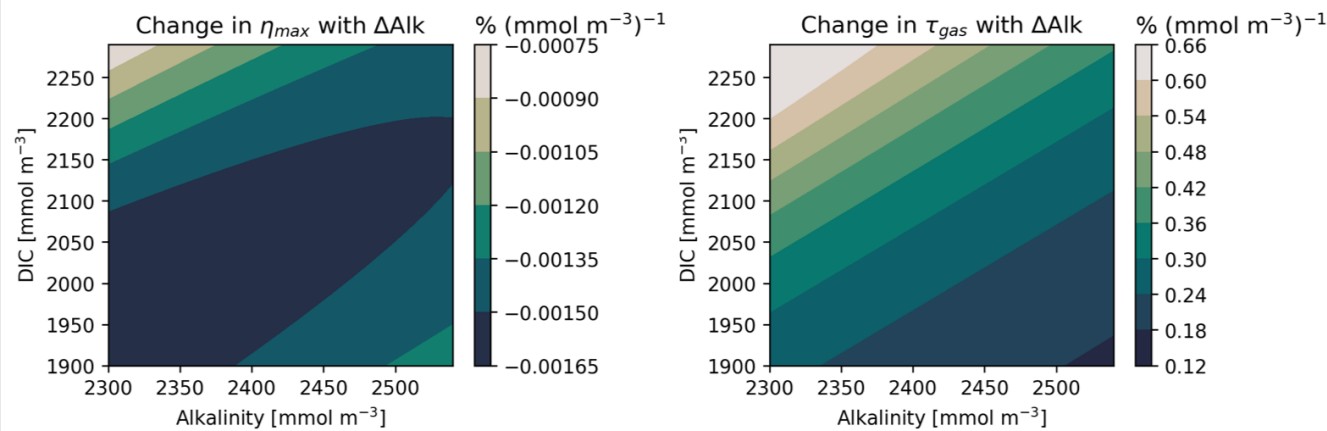

**Figure 4.** We assess how the efficiency $\eta_{max}$ and carbon uptake timescale $\tau$ change with the magnitude of the alkalinity perturbation $\Delta$Alk. We compute $\eta_{max}$ and $\kappa$ (Equation 7) for typical oceanic DIC and Alk values and perturbations $\Delta$Alk $= 1$ to $100$ mmol/m$^3$ with intervals of $10$ mmol/m$^3$. We find the linearized rate of change of $\eta_{max}$ and $\kappa$ relative to $\Delta$Alk. We may thus obtain the percent change in $\eta_{max}$ (left) and $\tau$ (right) per mmol/m$^3$ of alkalinity added.

a similar test for the gas exchange timescale, $\tau$. We assume that $K_0$, $h$, and $k_w$ are not functions of alkalinity, and thus the nonlinearity may be encapsulated by a parameter $\kappa$:

$$\kappa = \frac{\partial \mathrm{DIC}}{\partial p\mathrm{CO}_2}. \tag{7}$$

In the right panel of Figure 4 we compute the rate of change in $\kappa$ per mmol/m$^3$ of added Alk between $\Delta$Alk $= 1$ and $100$ mmol/m$^3$. We note that $\kappa$, and thus $\tau$, is more sensitive to the magnitude of the alkalinity perturbation than $\eta_{max}$ was, increasing up to $65\%$ for a $100$ mmol/m$^3$ Alk perturbation. Still, for the forcing values considered here we expect $\tau$ to increase by at most only a few percent, not significantly hindering the IRF linearity requirement (though this will be assessed in the results).

We acknowledge other potential sources of chemical nonlinearity may arise when considering the "additionality problem"
(Bach, 2024). The climatic benefit of OAE may be defined by its additionality, or how much the OAE intervention increases CO$_2$ removal relative to a baseline state without OAE. However, OAE may also modify the natural alkalinity cycle and associated baseline calcium carbonate saturation state, leading to changes in biogenic calcification. The study by Bach (2024) presents experiments showing that OAE can reduce the generation of natural alkalinity, thereby reducing additionality in many of the marine systems where OAE is being considered. The additionality problem is enhanced in natural alkalinity cycling
hotspots, such as marine sediments. Regarding time invariance, the carbonate chemistry equations are invariant by formulation. The only place where time invariance may be violated in the above equations is in $k_w$. Since $k_w$ is determined by wind speed, and thus the dynamical ocean/atmosphere component, we return to it in the next section on physics. Thus, from a purely chemical perspective, we have shown that the LTI requirement is a reasonable approximation.



## 3.2 Physics and Time Invariance

The physical equations governing the ocean and atmosphere system are not, to first order, functions of alkalinity, and we assume that the $\Delta$Alk perturbation magnitude will have a negligible influence on resulting dynamics for at least the decadal timescale following the OAE intervention. A caveat is that in our numerical simulations we assume a non-responsive atmosphere; i.e., the change in the atmospheric $CO_2$ reservoir is negligible. Although OAE will decrease the atmospheric $CO_2$ and thus impact $CO_2$ uptake, the atmospheric reservoir is significantly larger than the $CO_2$ perturbations considered here. Preliminary analysis

indicates that for carbon accounting purposes, the response of the atmospheric $CO_2$ reservoir may be neglected when assessing OAE influences. We assume that the linearity condition is sufficiently satisfied from a physical perspective. However, the time invariance condition poses ambiguity as to whether and how we may successfully apply IRFs for quantifying OAE-based CDR. Particularly for an OAE deployment occurring over a small spatial and temporal scale, the resulting carbon uptake curve may be significantly affected by local tidal flows, turbulence and mixing patterns, and the background seasonal and interannual

flow variability characterizing the particular time and place of interest. Weather patterns and resulting wind dynamics also strongly influence air-sea gas exchange, as $k_w$ in Equation 6 depends on the average square of 10-meter height neutral winds (Ho et al., 2006; Wanninkhof, 2014). Similarly to physical subduction, wind speed variability may lead to drastically different carbon uptake curves and violate time invariance. We employ two strategies to deal with these challenges posed by ocean and atmosphere variability. First, for our IRF experiments we use alkalinity pulses that extend for a one-month duration and

span several hundred square kilometers, sufficient to integrate over dominant periodicities in local flows (e.g. tides or eddy features). Additionally, we generate libraries of IRFs capturing seasonality and perform ensembles to estimate the uncertainty associated with interannual variability. Nonetheless, due to the complexity and nonlinearity of the ocean-atmosphere system, it is challenging to say a priori how this will project onto our ability to utilize IRFs for CDR quantification successfully.

## 3.3 Biology

The biological system of the ocean poses a challenge to the LTI requirement due to its inherent complexity and potential for feedbacks. Bach et al. (2019) provides a survey of knowledge gaps and hypotheses regarding how different chemical forms of OAE may impact the biological system. For example, in OAE interventions employing quicklime (CaO), calcifying organisms are expected to benefit. On the contrary, when using olivine, silicifiers and cyanobacteria are expected to increase primary productivity significantly. Either scenario may lead to changes in the biological carbon pump. Suessle et al. (2023)

discuss how the vertical fluxes of carbon, nitrogen, phosphorus, and silicon may be affected by perturbation magnitudes of alkalinity. An encouraging result of their work is that carbon export by oligotrophic plankton communities is insensitive to OAE perturbations (though they suggest additional investigation in less idealized models). In the last year, numerous studies have probed the influences of various marine-based CDR approaches on biological and carbon cycle feedbacks. For example, Berger et al. (2023) found that when macronutrient limitations and biological feedbacks are parameterized in their model, the

efficiency of macroalgae-based CDR drops. Other studies have considered OAE effects on coastal fish larvae (Goldenberg et al., 2024), diatom silicification (Ferderer et al., 2023) and phytoplankton responses (Xin et al., 2024; Hutchins et al., 2023).



We conclude that assessing a priori the magnitude of nonlinearity and time invariance of the biological system to perturbations in $\Delta\mathrm{Alk}$ is beyond the scope of this work. We acknowledge the potential for biological feedbacks and the importance of considering and quantifying such phenomena. In the near term, however, we expect OAE signals to be small except in the local area immediately adjacent to perturbations (the near-field). We envision an approach employing observations and high-resolution models to quantify the near-field dynamics, including various feedback elements, and subsequently hand off these results to an IRF-based approach at larger scales where signals are more diffuse.

## 4 Numerical Experiments

Having presented the IRF framework and the physical, chemical, and biological system constraints to applying IRFs in the real ocean, we move towards applying IRFs for quantifying OAE-based CDR in a global climate model. The objective is to first obtain an IRF (or a "library" of IRFs accounting for seasonal variability) and then use it to predict the uptake curve of a continuous alkalinity deployment. We will assess the accuracy with which the IRF prediction can reproduce the actual continuous release model simulation, and consider several distinct dynamical regimes in the ocean with differing biogeochemical properties and flow dynamics.

Our study followed the experimental design outlined in Zhou et al. (2024), hereafter referred to as Z24. Z24 employed the Community Earth System Model version 2, CESM2 (Danabasoglu et al., 2020), in the forced ocean-ice configuration, FOSI (Yeager et al., 2022). The ocean component of the model is the Parallel Ocean Program version 2 (POP2), integrated at a 1° horizontal resolution and forced by the Japanese 55-year atmospheric reanalysis dataset, JRA55 (Kobayashi et al., 2015). The forcing includes the historical transient in atmospheric $CO_2$, and the model was integrated using repeating JRA55 forcing cycles from 1850 to near-present, thus accounting for the accumulation of anthropogenic $CO_2$ in the ocean. POP2 is coupled to the Marine Biogeochemistry Library, MARBL (Long et al., 2021). MARBL simulates two carbonate systems in parallel online; thus, the baseline and the case with the OAE intervention are computed simultaneously within a single model integration. The Z24 study divided the Global Ocean into 690 polygons and performed separate global model integrations for OAE deployments within each polygon. Alkalinity was released at the polygon's surface at a constant rate of 10 mol m$^{-2}$ yr$^{-1}$ for one month, and the model was integrated for 15 years post-release. As discussed in Z24 and He and Tyka (2023), such a flux magnitude generally ensures that: (1) the pH increase is less than 0.1, which is the amount that pH has already decreased in the surface ocean since preindustrial times (Doney et al., 2009), and (2) the aragonite saturation state change is less than 0.5, preventing the secondary precipitation of calcium carbonate which would decrease OAE efficiency (Moras et al., 2022). To investigate the seasonal variations of OAE efficiency, four separate simulations were performed for each polygon with alkalinity releases in January, April, July, and October 1999. Thus, Z24 provides the first geographically-refined global map of OAE efficiency with an assessment of seasonal variability. For additional details on the experimental design, model details, and discussion of global variations in OAE efficiency, please refer to Z24 (Zhou et al., 2024).

In our work, we treated the simulations of Z24 as the "impulse" experiments. Each of their OAE deployments lasted one month, which is sufficiently short to resolve seasonal and interannual variability. We first identified several polygons corre-





245    sponding to different dynamical regimes and consider their uptake curves obtained by Z24 as the "IRFs". We chose 9 polygons
in the North Atlantic, and 8 polygons in the North Pacific, with various efficiencies and seasonal variability as found by Z24.
We then performed continuous OAE release experiments beginning in 1999 and lasting five years, with the same magnitude
of forcing as in Z24. For select polygons, we also performed a one-year continuous release experiment. We computed a con-
volution of our prescribed forcing with a seasonal library of uptake curves obtained in Z24 to get an IRF-based prediction of

250    the carbon uptake efficiency as a function of time for the continuous release simulations. We then compared the explicitly-
simulated and predicted efficiency curves to assess the validity of the IRF approach. To quantify the uncertainty associated
with interannual variability, we additionally performed ensemble simulations for a selection of the polygons we considered.
The ensembles consist of separate one-month alkalinity deployments (identical to Z24) in January of 16 years, ranging from
1999 to 2014. Examining the spread in uptake efficiency curves across these 16 January OAE deployments provides insight

into the expected uncertainty of the IRF prediction (which is based on just one year) for a given polygon.

## 5    Results

Here, we go through the procedure for applying the IRF-based CDR quantification to the OAE problem. We will begin by
discussing the spatiotemporal scales of the OAE problem, discuss seasonal and interannual variability and its influences on
the IRF approach and resultant uncertainty, and end by showing a synthesis evaluating the performance of the IRF prediction

across different ocean basins and dynamical regimes.

### 5.1    Spatiotemporal Scales

The first step in applying the IRF methodology is to obtain the IRFs, based on the month-long OAE release simulations of Z24.
Figure 5 presents one such simulation. Here, alkalinity was released in January 1999 off the coast of southern California, and
the model was integrated for 15 years post-release. The excess alkalinity at the surface is seen to spread through the entire zonal

extent of the Pacific by about 10 years. By 15 years, most of the surface of the Pacific within 40° of the equator has excess
alkalinity associated with the intervention.  In Figure 6 we show the distribution of excess alkalinity at the surface for the
January "impulse" simulation as well as our two continuous release cases, 15 years post-release. One can see that the surface
alkalinity distribution, when normalized by the total amount of alkalinity added (done by scaling the colorbar by a factor of 12
and 60 for the 1- and 5-year releases, respectively), appears to have the same distribution for all cases. The lower left panel of

Figure 6 shows the time series of alkalinity forcing – note that alkalinity is put into the surface ocean at the same rate of 10
mol m$^{-2}$ yr$^{-2}$ for the four month-long releases and the continuous releases. The middle panel shows the net change in DIC
inventory for each case, and the last panel shows the curves of OAE efficiency $\eta$ for each case. Note that the blue curves are
our "IRFs" since they represent the response of the system to an alkalinity forcing impulse. The continuous-release simulations
have different uptake curves; note that the kinks in the curve represent the cessation of alkalinity addition.





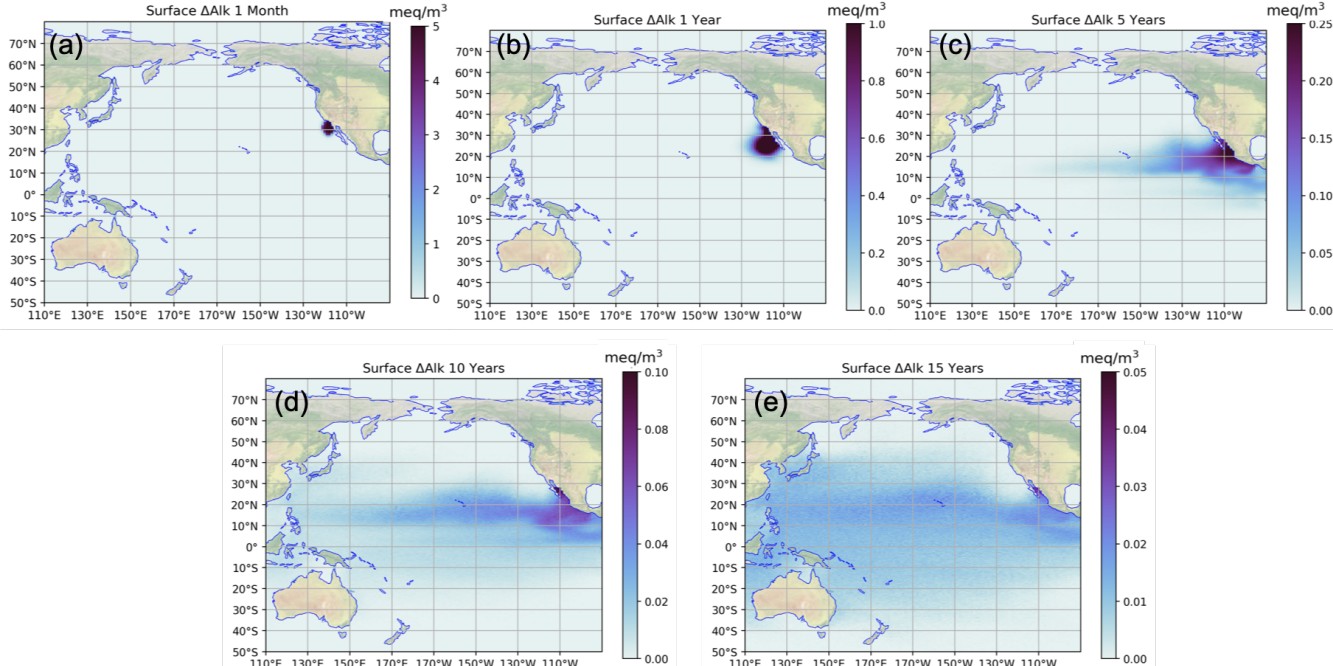

**Figure 5.** Evolution of excess alkalinity at the surface following the month-long January alkalinity release off the coast of California, see Polygon 214 in Figure 12. Times post-release are 1 month, 1 year, 5 years, 10 years, and 15 years (a-e, respectively).

## 5.2 Seasonality

As discussed above, our metric for the IRFs is efficiency $\eta(t)$. Z24 provides us with four IRFs corresponding to different seasons, but a few details should be noted regarding their implementation. Since our IRFs end at 15 years, we fit an exponential curve to them so that we have an analytical form for the IRF that we may extend longer in time. The exponential has the form (Zhou et al., 2024):

$$h(t) = \eta(t) = \eta_{max}[1 - \alpha e^{-t\left(\frac{1}{\tau_1} + \frac{1}{\tau_2}\right)} - (1-\alpha)e^{-\frac{t}{\tau_3}}], \text{ with } \alpha = \frac{\tau_2}{\tau_1 + \tau_3}. \tag{8}$$

In Equation 8, $\tau_1$, $\tau_2$, and $\tau_3$ are adjustment timescales over which the $CO_2$ equilibration occurs; $\tau_1$ is the fast adjustment driven by gas exchange, $\tau_2$ is the longer-term adjustment, and $\tau_3$ represents the transition from $\tau_1$ to $\tau_2$. We found this analytical form to reconstruct the IRF data successfully. Note that we could just as readily use the $\eta(t)$ curves themselves without performing the fitting, but to decrease the data volume and allow for longer time series in our convolutions, we opted for the curve-fitting route. The next detail is how to apply the IRFs to perform the convolution with forcing and obtain an IRF-based carbon uptake prediction. Since our two continuous release simulations both begin in January, our first approach is simply to take the January IRF and proceed. However, Z24 identified many regions as having strong seasonality. This means that our time invariance condition is violated – the IRF changes substantially based on the season. In Figure 7, we perform a linear interpolation of





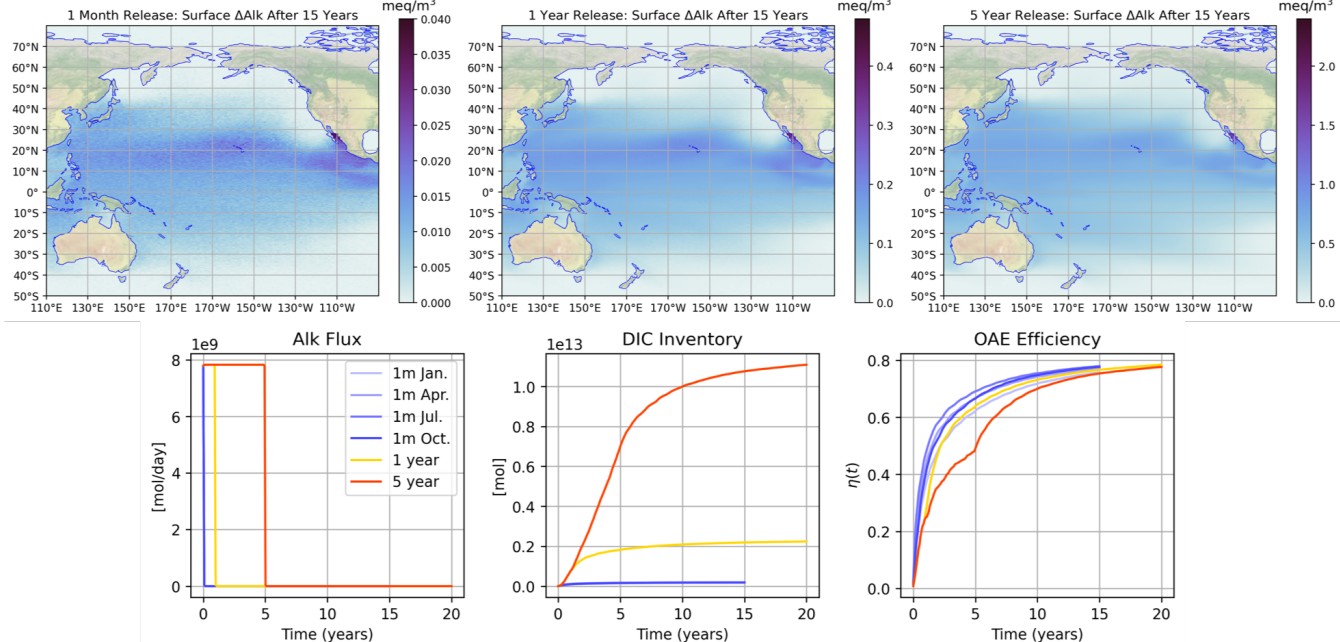

**Figure 6.** Overview of the simulations used to test the IRF approach at the southern California location (Polygon 214 in Figure 12). The top row shows the excess alkalinity at the surface for the month-long January OAE simulation 15 years post-release, the 1-year continuous release 15 years post-release (16 years since the beginning of the model integration), and the 5-year continuous release also 15 years post-release. The colorbars are adjusted for the latter two based on the total amount of alkalinity released so that they are directly comparable to the first plot in terms of color. One can thus observe that the surface alkalinity distributions between these cases are nearly identical. The bottom row shows the alkalinity forcing time series for the four separate month-long and two continuous release cases, the excess DIC inventory as a function of time, and the OAE efficiency ($\eta$) as a function of time for each case.

the January, April, July, and October 1999 IRFs from Z24 to get a month-by-month IRF. We show the resulting IRFs for the
California location considered in Figures 5-6. Note that at this location, all IRFs are relatively close together, i.e., seasonality is weak, and time invariance is not violated significantly. However, considering the same plot for the Iceland OAE location, we notice up to a 150% increase in efficiency between the winter and summer months (Fig. 7b).

Our first approach is to simply take the January IRF and perform the convolution with a time series of 1-year and 5-year continuous alkalinity forcing to obtain the resulting IRF predictions, $\eta(t)_{1yr,5yr} = \Delta\text{Alk}(t) * \eta(t)_{Jan}$. We compare the actual
and predicted $\eta(t)$ for the continuous release simulations in Figure 8. For the California location, the January IRF prediction performs well and aligns with the $\eta(t)$ curves for both continuous-release simulations. The greatest deviation occurs in years 2-5 and is attributable to interannual variability, but after a few years, the predicted and actual curves converge. However, this is not the case for Iceland. We see a significant underestimation of about 20% by the IRF compared to the model result. This is because we have violated time invariance and are using the January IRF, which does not encapsulate the summer behavior
at this location. So, instead, we perform the convolution month-by-month using the monthly IRFs from Figure 7. Note that at



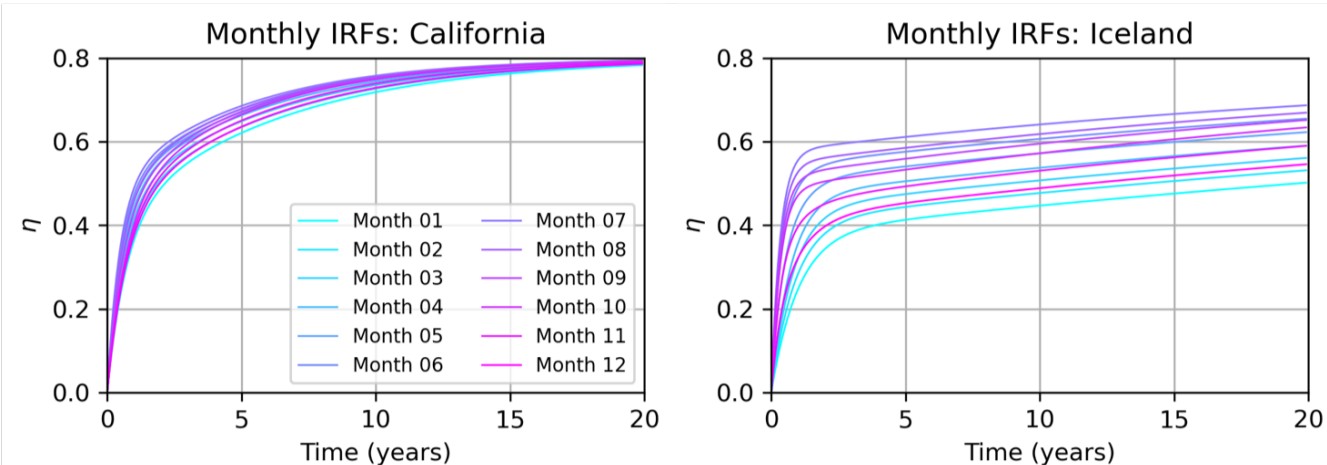

**Figure 7.** Using the month-long alkalinity release simulations conducted in January, April, July, and October by Zhou et al. (2024), we construct a library of IRFs for each location corresponding to each month by linearly interpolating between months 1, 4, 7, and 10. The left panel shows the resulting monthly IRFs for California (Polygon 214, Figure 12), and the right panel is Iceland (Polygon 36).

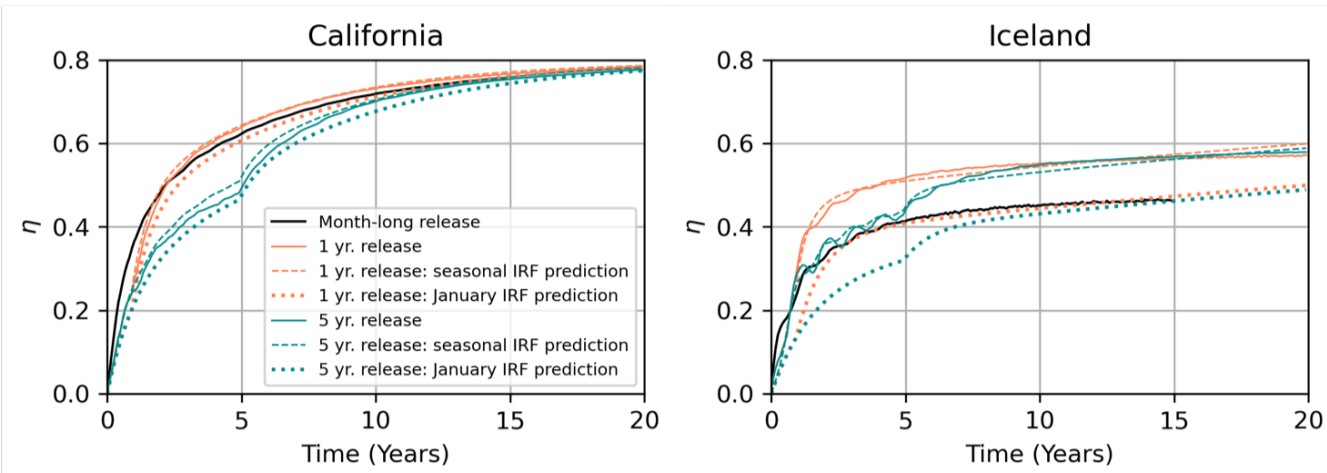

**Figure 8.** Comparison of the IRF prediction and model integration result for the continuous release simulations in California (left) and Iceland (right), Polygons 214 and 36 in Figure 12. The black line shows the OAE efficiency ($\eta$) curve for the month-long January release. The solid orange and dark cyan lines are the OAE curves obtained from the model integration of a 1-year and 5-year continuous alkalinity release experiment, respectively. The dashed line shows the corresponding prediction using IRFs constructed for each month as shown in Figure 7. The dotted line is the prediction using only the January IRF, without constructing an individual IRF for each month to account for seasonal variability.





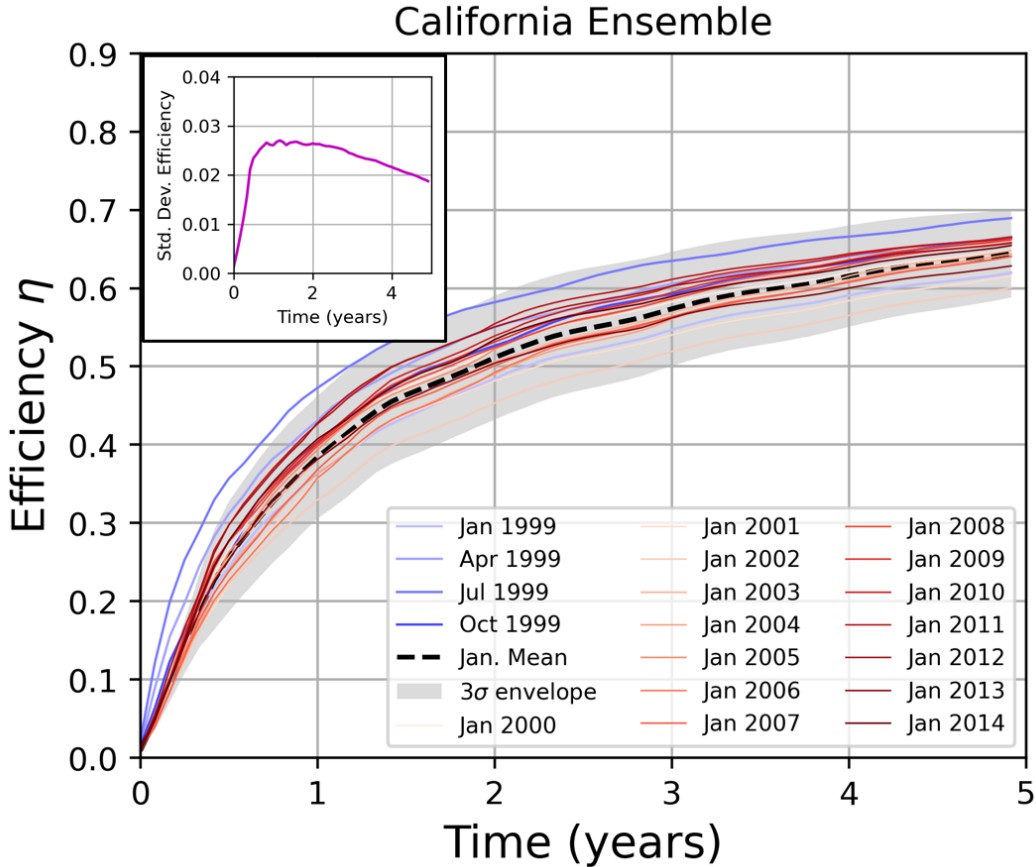

**Figure 9.** Ensemble results of efficiency $\eta$ for the California location (Polygon 214, Figure 12). We conduct a series of 16 model integrations beginning in each year from 1999 to 2014, with a month-long release of alkalinity in January of each starting year. These results are shown as red lines, and their mean is shown as a dashed black line. In gray is the envelope encompassing three standard deviations from the mean; in blue are the four seasonal simulations of month-long OAE deployments in January, April, July, and October of 1999. This figure allows one to compare the seasonal variability in $\eta$ to the interannual variability. The insert shows the standard deviation in $\eta$ of the January releases at this location as a function of time. Analogous plots for other locations are shown in Figure 11.

the Iceland location, using the monthly IRFs substantially improves the IRF prediction, so it is nearly aligned with the actual model result. Interestingly, at the California location, the seasonal and January IRFs have roughly the same deviation from the actual curve. This indicates that the interannual and seasonal variability have similar magnitudes, whereas in Iceland, the seasonal variability substantially dominates in magnitude.



## 5.3 Interannual variability

To gain deeper insight into the variability — particularly to test our statement above that seasonal and interannual variability are comparable at the California location — we performed ensemble simulations detailed in Section 4. This is illustrated in Figure 9 for the California location. One can immediately observe that there is variation in January uptake curves depending on the year and, as predicted, the variability is comparable to the magnitude of the seasonal variability. Interestingly, the standard deviation of the ensemble members peaks between 1-2 years and then decreases. This explains why in Figure 8, the 1999-based IRF prediction deviates slightly from the explicit simulation in the several years following the OAE deployment and then converges to the model result.

We have shown that the IRF-based CDR quantification is valid once the time variance of the IRF is accounted for at two sample locations – California and Iceland. We now test the approach of applying the monthly IRFs capturing seasonal variability at four other locations with differing dynamical regimes and uptake curves. These regions are: the Labrador Sea, North Sea, Western Equatorial Pacific, and North Pacific (see Polygons 0, 16, 334, and 305, respectively in Figure 12). As in the prior section, we first compute a monthly IRF for each location by fitting an exponential curve to the four seasonal IRFs from Z24 and linearly interpolating between the months. We compute the convolution of IRF and alkalinity forcing month-by-month for our 1-year and 5-year continuous alkalinity forcing cases. We perform model integrations for the continuous OAE releases and compare the resulting uptake with our IRF prediction. Finally, we perform ensembles consisting of 16 members. Each member simulates a month-long January alkalinity release in a year ranging from 1999-2014, and is integrated for five years post-release.

The results for these four locations are shown in Figure 10. We first note that the IRF prediction matches nearly perfectly with the 1-year release simulation at all four locations. This agreement stems from the fact that the monthly IRFs used in this prediction were taken from the year the alkalinity was released, so we mostly account for the effects of seasonal variability— and interannual variability is not a factor. However, for the 5-year case, we are still using the 1999 IRF, even though alkalinity continued to be released from 1999-2004. As a result, the IRF does not capture the effects of interannual variability. Nonetheless, we see remarkable agreement between the IRF and model results in all regions except the Labrador Sea (where there is roughly a 10% error). In the North Sea, interannual variability becomes important 2-6 years into the simulation, and the IRF diverges from the model result. However, over time, the ensemble spread decreases (variability averages out), and the IRF curve converges with the explicit simulation results after 7 years. Similar, though slightly smaller trends, are also apparent in the Pacific cases. The interannual variability is shown in Figure 11. For the Labrador Sea, the model simulates a large interannual variability that explains the error in the IRF prediction. At this location, strong wintertime convection removes the alkalinity from the surface and leads to very low values of $\eta$. The magnitude of this convection is highly variable year-to-year. We note that the year 1999, for which our IRFs were constructed, had particularly strong convection and a lower-than-average efficiency – this is why the IRF prediction in Figure 10 underestimated the actual curve. Furthermore, the standard deviation of the ensemble members increases as a function of time, so the IRF and model results do not converge with time, implying that convection removes alkalinity from the surface for longer timescales than the 10-20 years we consider. Interestingly, at the





**Figure 10.** Comparison of the IRF prediction and model integration results for the continuous release simulations in the Labrador Sea (Polygon 0, Figure 12), North Sea (Polygon 16), the western equatorial Pacific (Polygon 334), and north Pacific (Polygon 305). The black line shows the OAE efficiency ($\eta$) curve for the month-long January release. The solid orange and dark cyan lines are the OAE curves obtained from the model integration of a 1-year and 5-year continuous alkalinity release experiment, respectively. The dashed line shows the corresponding prediction using IRFs constructed for each month.



**Figure 11.** Ensemble results of efficiency $\eta$ for the same locations as Figure 10. Month-long January releases from 1999 to 2014 are shown as red lines and the mean is shown as a dashed black line. We also show the envelope encompassing three standard deviations from the mean, as well as the four seasonal simulations of month-long OAE deployments in 1999 (blue lines). The lowest row shows the standard deviation in $\eta$ of the January releases as a function of time.





**Figure 12.** (a) Polygons defined by Zhou et al. (2024), with the ones considered in this study for testing the IRF approach labeled in black. (b) Bar charts show the ratio of the modeled continuous release compared to the IRF prediction 15 years after the end of the alkalinity release. Dark pink is the 5-year continuous release case, and violet (semi-transparent) is the 1-year continuous release case. The numbers on the bars indicate the standard deviation after 5 years of ensembles performed for a given polygon. Note that not all polygons have an ensemble result, and not all polygons have a 1-year result (due to computing constraints). (c) Similar to (b), here we present the IRF prediction vs. the actual model result for the 5-year release cases, with the one-to-one line shown as a black dashed line. Error bars show the standard deviation from the ensemble results.

340 three other locations, the ensemble standard deviation peaks and then decreases. Again, this explains why the IRF predictions that initially deviated from the model result eventually converged. Interestingly, despite the interannual variability often being similar in magnitude to seasonal variability, and our IRFs not taking into account the interannual variance, we still obtain an excellent matchup with the model results.



## 5.4 Comparison across regions

In Figure 12 we synthesize all of the results for the polygons considered in this study. The top panel shows a map with labeled
polygons for which we performed continuous release simulations to test the IRF-based CDR quantification. Note that the 1-year continuous release case is less "challenging" for the IRF approach as our IRFs were also constructed from 1999 impulse experiments. The 5-year case faces the time invariance problem, since we're adding alkalinity to years other than 1999 but still using the 1999 IRFs. The bar charts in panel (b) show the ratio of actual to IRF-predicted OAE efficiency $\eta$ 15 years post the OAE release. Remarkably, in all cases, the error of the IRF approach is less than 7%, even in the highly variable Labrador Sea. In the Pacific, the errors are even smaller, within 1% of the model result. We additionally label each bar with the standard deviation of the ensemble performed at that polygon location. One can see that the higher errors stem from the higher interannual variability (violating the time invariance requirement). In panel (c) we create error bars using a single standard deviation $\sigma$. The 1:1 line shows where the IRF prediction matches the actual model result, and nearly all points with the associated error bars fall onto this line. We conclude that once an appropriate seasonally-varying IRF is constructed, the IRF approach may be applied to successfully predict the carbon uptake associated with an OAE deployment without the need to perform a model integration explicitly. We are able to stay within sufficient bounds of linearity and time invariance required for the IRF approach to work. Additionally, we note the potential of using an ensemble of IRFs over the historical period to account for interannual variability.

## 6 Discussion and Conclusions

We have presented compelling results indicating that impulse response functions (IRFs) may be a powerful tool in aiding the Monitoring, Reporting, and Verification (MRV) of ocean alkalinity enhancement (OAE) interventions. The premise is that we can probe the complex ocean, terrestrial, biological, and atmospheric system by forcing it with an "impulse" – a short pulse of alkalinity applied to the surface ocean. By constructing an OAE efficiency curve for the resultant carbon uptake from our intervention, we can predict the carbon uptake for any arbitrary time series of alkalinity forcing, provided we have a linear and time invariant (LTI) system. We first discussed and quantified the sources of nonlinearity and time invariance in the system. We then tested the IRF approach in the CESM 1-degree resolution global climate model. For our impulse simulations, we used the OAE Efficiency Atlas created by Zhou et al. (2024) (Z24). We performed continuous alkalinity release simulations, and found remarkable agreement between the IRF prediction and model results. In most regions, the IRF prediction is typically within 1% of the actual model result. We determined the accuracy of the IRF to hinge predominantly on the magnitude of the interannual variability at the given location; we quantified this by performing 16-member ensembles.

Though this study has shown great potential for future use of IRFs for MRV of ocean alkalinity enhancement interventions, many additional questions have come up. The biggest challenge in assessing OAE influences is its multiscale nature, which is highly sensitive to the various modes of variability comprising the Earth system. The global model used here has a horizontal resolution of 1 degree ($\approx$ 100 km). This resolution cannot resolve oceanic mesoscale features, such as eddies, or submesoscale dynamics such as fronts, filaments, mixed layer eddies, and inertial-symmetric instabilities. These smaller scale dynamics may



lead to vertical transport of alkalinity that could be highly relevant to setting the OAE efficiency. There is also the question of how realistic our initial condition and forcing scenario are. Our polygons were all on the order of several hundred kilometers in length and width and experienced constant alkalinity forcing over that area with a relatively small flux of 10 mol m$^{-2}$ yr$^{-1}$. In practice, alkalinity will be released from point sources, and tracing its progression from the order of a few meters to the size of the polygons represents a large technical challenge. Due to the multiscale nature, there is no numerical framework capable of resolving all of the relevant spatiotemporal scales and multi-scale coupling is highly computationally expensive.

Constraining the small-scale turbulence and mixing influencing the near-field and short-term evolution of an alkalinity plume and coupling this to the larger-scale dynamics remains an important avenue for future work. We envision using the IRF for tracing the evolution of the OAE intervention at the larger, gas-exchange scales. Linking the high-resolution near-field modeling/observations to the coarser-resolution regional/global scale modeling remains a critical component to successful MRV of OAE. Additionally, exploring what length of "impulse" is appropriate (here we only considered one month) may be an interesting question to pursue when considering various modes of oceanic variability. In our coarse global model, the month-long pulses were sufficiently short compared to the characteristic seasonal variability. As a result, we constructed a library of IRFs sampling over this variability. If we consider high-resolution simulations of a tidally-driven estuary, for example, an hour-long pulse would preference a particular tidal phase, and we would similarly need to construct a library of IRFs sampling the tidal cycle. Or, we could envision a longer deployment of 12–24 hours to sample over the full tidal cycle. Another assumption in our work is that of non-interactive atmosphere and terrestrial components. Although generally considered second-order effects for the purpose of carbon accounting (Tyka, 2024), interactive atmosphere and terrestrial carbon pools may be important future considerations. An interactive atmosphere decreases the sensitivity of the biological pump to changes in carbon uptake, and including the terrestrial component increases the fertilization-induced marine carbon uptake Oschlies (2009). Exploring a more realistic interaction of the ocean, atmosphere, and land systems is an important future step in understanding OAE influences in projecting onto the global carbon budget (Friedlingstein et al., 2023).

The next steps in this work involve closing the gap between the laminar dynamics of the order 100 km grid model used in this study and the meter-to-kilometer scale dynamics potentially relevant for MRV of OAE interventions. We are applying the results of this work to guide further regional modeling efforts aimed at resolving mesoscale turbulence, and potentially some submesoscale dynamics. We anticipate future work applying the Regional Ocean Modeling System (ROMS) (Shchepetkin and McWilliams, 2005) to perform high-resolution regional simulations analogous to the CESM simulations presented here. Regional simulations will allow us to study the role of smaller-scale ocean dynamics in modulating carbon uptake and OAE efficiency. A prior study by Wang et al. (2023) performed an OAE simulation in the Bering Sea using a 10 km resolution ROMS configuration. They emphasize the need for modeling at various scales and their study helps set the stage for our future research. We hope to address fundamental questions of whether/how OAE dynamics change as a function of model grid resolution. We will also assess the performance of the IRF prediction as increasing flow variability becomes resolved. Overall, based on these promising results, we believe the IRF approach provides a foundation for quantification and uncertainty assessment of OAE.



*Code and data availability.* The experiments and data used to construct the impulse response functions detailed in the manuscript are pre-
410  sented in Zhou et al. (2024). A repository showing the IRF methodology may be found at: https://github.com/CWorthy-ocean/IRF_Method.
Jupyter notebooks used to analyze the data and create the figures in this manuscript may be found in the Zenodo repository: https://doi.org/
10.5281/zenodo.13392377.

*Author contributions.* ML conceived the idea underpinning this work. EY, SB, and MZ performed the model simulations. EY, SB, AK, and
ML developed the IRF methodology and analysis used in this study. All authors contributed to data analysis, interpretation, and discussion.
415  All authors participated in the writing and editing of the manuscript.

*Competing interests.* No competing interests are present.

*Acknowledgements.* We thank [C]Worthy team members Dafydd Stephenson, Thomas Nicholas, Ulla Heede, Namy Barnett, and Toby
Koffman for insightful discussions during the course of this project. The authors acknowledge high performance computing support from the
Cheyenne supercomputer (https://doi.org/10.5065/D6RX99HX). Any opinions, findings, conclusions, or recommendations expressed in this
420  material do not necessarily reflect the views of the National Science Foundation.



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
