# Peer review of "Impulse response functions as a framework for quantifying ocean-based carbon dioxide removal"

_EGUsphere, 2024_

## Author Response (AR1)

**Reviewer 1:**

The authors have explored nuances of a possible path forward that the community has been collectively been considering for model-based MRV for marine CDR: impulse response functions. IRFs can be used to estimate efficiency losses from incomplete air-sea gas exchange following an intervention. The authors present some mathematics and schematics explaining how this might be able to be done. They do this using a model that captures some seasonal and interannual variability in ways that alternative approaches to deriving IRFs (e.g., from transport matrices) cannot.

My main criticism for this manuscript is that it addresses a "cat" in the room rather than the "elephants." Specifically, the questions that are addressed by the model are:

- 1. Can an IRF be effectively discretized using 4 distributed seasonally-specific IRFs? (This question is adequately addressed.)
- 2. To what degree does the IRF break down due to interannual variability as the length of time between its definition and use grows? (This question is not well addressed or posed, but some of the results speak to the issue.) We have added a paragraph on this question (Section 5.4).

The authors do not address the main problems with IRFs, which swirl around the question of whether the model is adequately representing the true Earth system (are the resolution; parameterizations of biological, atmospheric, and terrestrial feedbacks; parameterizations for mixing, parameterizations of gas exchange; initialization; and forcing adequate to resolve the signal of interest?). Normally that would be okay, and a nice paper can be written that does a targeted analysis of a limited question, except that the authors present their analysis as an assessment of the viability of IRFs for MRV generally. This makes the central argument of the paper feel a bit like a "strawman" argument. When a subset of these other issues are raised, they are mostly dismissed using the logic that (Paraphrased to make a point... this is not a quote) "We don't have to worry about some challenges to the IRF framework because they only become relevant if we do mCDR in a way that might affect the Earth system." If this is a fair argument, then the paper is making itself irrelevant by arguing that these approaches to mCDR are not viable at a meaningful scale. It would be better if these issues were more quickly brought up and listed as issues that are not addressed at present rather than presented as issues that can be dismissed.

Another limitation of the paper is that it seems to rely on access to IRFs that are specific to a both a location and a time of release. Several recent studies have blanketed the global surface ocean with IRF estimates and the great Zhou et al. study referenced indeed provides seasonally varying global IRFs. However, it seems unlikely that most people using IRFs will have estimates that are specific to the same year as the release, as is assumed in this study. It would therefore be helpful if the authors could use their analysis to propose a more quantitative approach for assessing uncertainty in IRFs when they are used in different years from when they are determined (or better yet, from another year in another model entirely). The author's results speak to interannual variability, but the presentation feels anecdotal and doesn't provide actionable recommendations for quantifying this uncertainty at a general location.

We first thank the reviewer for the detailed assessment of the manuscript and the line-by-line suggestions; we have implemented many of these ideas into our revisions. We entirely agree with the need for model validation and have added discussion of the potential biases the 1-degree model may introduce into the IRF approach as well as model limitations (see Discussion and

Conclusions as well as other additions throughout the text). Numerous literature has been published that employs NCAR's CESM 1 degree model, including Zhou et al. 2024. We thus do not believe that our manuscript is a place to do a detailed validation of the model but agree that there is a need to address the model's performance and limitations in the context of the IRF methodology. We have done this by citing appropriate literature and speaking about the gaps in resolving mesoscale/submesoscale turbulence (which may present challenges to implementing IRFs). We are performing additional investigations into these questions using higher-resolution regional models, and speak about this in the manuscript. The resolution dependence of uptake efficiency and variability is a deeper question that requires additional research efforts. We view a large part of this paper's contribution as a proof-of-concept of using IRFs within a given modeling framework (with potential to generalize to other models or model inter-comparison projects).

Regarding the dismissal of some challenges as not being relevant until mCDR is done on a larger scale: the scaling of the mCDR industry must pass through a period where deployments are small in scale, but still require robust verification to support transactions of carbon removal. We believe IRFs may have a role to play here. The main place where this is done is in assuming a "noninteractive" atmosphere, i.e. the pCO2 in the atmosphere doesn't change for the duration of the OAE simulation. This is a simplification but we have cited literature (Tyka 2024) that addresses this by performing simulations with and without an interactive atmosphere (we did so earlier in the paper as requested). The IRF curves presented here, as well as the results by Zhou et al. 2024, are concerned with the intrinsic OAE efficiency, i.e. relative to a direct atmospheric removal (such as DAC) of the same magnitude. It has been shown (Tyka 2024) that the use of a prescribed atmosphere yields an efficiency metric which only measures this relative efficiency and that is what the present paper is concerned with. On the other hand, the absolute efficiency, meaning the reduction of atmospheric CO2 following some intervention, is obtained when using responsive atmospheres or earth system models. However, this conflates the intrinsic efficiency with the effects of any negative or positive emissions on the rebalancing of CO2 inventories between various interconnected reservoirs (e.g. Ocean, Atmosphere and Terrestrial biosphere). As these are common to all negative emission technologies, they are not addressed here. If there are other specific places where we have inadequately spoken about simplifications we will be happy to explain in more detail.

Regarding the last paragraph of the above text: we have presented quantification of interannual variability for numerous locations in which we tested the IRF methodology. Our 5-year releases involve releasing alkalinity for 5 years and using an IRF that was developed for only the first year (we quantified how well these perform and in most regions they do well). Please see Section 5.3. We have added an additional paragraph on the longer-term variability and some thoughts on how to apply the IRFs incorporating knowledge of the interannual variability (please also see Reviewer 3's comments). Please see the edited Section 5.4.

I also would criticize the presentation on two accounts: the complexity and the organization.

Regarding complexity: The main ideas of the paper are reasonably simple, but they are presented in unnecessarily complicated ways. Admittedly, for a formal MRV approach it is important to explicitly present every step in a calculation and drill down into the details to make certain that the calculations are being done in ways that are both practical and defensible. This paper therefore has appropriate ambitions despite the simplicity of the underlying calculations provided the authors can make the underlying math both exactingly correct and highly accessible. I worry that at present the manuscript seems to do neither, and manages to make the simple math behind the proposed idea appear complicated. The presented math also seems to have errors, as noted by an earlier public comment. A solution to both would be to keep the summation notation rather

than switching to an integral notation and just accepting that any OAE intervention can be approximated as a sequence of discrete releases rather than an infinite number of infinitesimal releases. This seems likely to be how the IRFs will be implemented in practice in any event, and it seems strange to worry about discretizing the release much finer than the 4x/year IRF functions that will need to be interpolated. Indeed, it is not clear that further discretization of the releases beyond 4x 3-month-long releases could even potentially result in any disagreement whatsoever from the instance with 4x 3-month-long releases because the authors have only modeled releases with no temporal variability between the beginning and end of the release (except in one schematic, which implies a great deal of short-timescale variability). It is possible that linear interpolation of response functions results in a non-linear effective response function, but that math wasn't fully explored.

We have corrected the errors pointed out in the review comments (we thank the reviewers for catching those), and have rewritten and streamlined the mathematical presentation to be clearer.

Regarding organization: The treatment of the various issues that are not addressed (aside from dismissal as mentioned above) and the introduction of the natural "thermostat" hypothesis are mentioned at an unhappy medium level of detail: just enough to convince the readers that these topics will be addressed by the manuscript, but not enough to address them. This makes the presentation somewhat confusing and longer than it needs to be. Similarly, there is some repetition of ideas (e.g., in captions and text) and scattering of methods text throughout the document that makes the paper longer than it needs to be.

In summary, this paper has a worthy if modest aim and good "ingredients" (that is, analysis and simulations). However, it is sufficiently miss-marketed and occasionally overstated that I believe it needs to be rewritten to be shorter, more focused, simpler, and more straightforward in its aims.

We appreciate the feedback and agree that the paper can be streamlined and more focused; we have revised the manuscript accordingly (it's still the same length but with numerous text additions addressing reviewer comments, and old content streamlined). Please see the more specific responses below. We somewhat disagree about "mis-marketing" and view the IRF approach as a promising proof-of-concept that requires additional research in models with increased turbulent variability. It does raise many challenging problems of how to handle the rich variability across a variety of temporal and spatial scales that profoundly affects carbon uptake (only part of which are captured in the model presented here). We have better caveated the limitations of using the 1-degree CESM in our revisions (Discussion and Conclusions section). However, once the challenging problem of encapsulating different forms of variability into an IRF library is accomplished, the IRF approach can greatly facilitate the MRV problem.

Line by line comments are transcribed as going through the paper for the first time... some questions raised in these comments are answered later in the manuscript, but are kept in these comments because they were questions or objections that should likely have been dealt with before that point in the paper.

We appreciate these comments and address them below; we also note that the ordering of some of the information presented in the paper is a matter of personal preference/style, and different readers will have different questions arise at various points in the manuscript. Our aim is to have all of the information present and logically ordered, and have made some modifications in the revisions to improve upon this.

1-5: why is the abstract focused on OAE when the title is not and the issues raised are not specific to OAE vs. e.g., DOC?

Please see Introduction: "Here, we develop the idea of using impulse response functions (IRFs) as a statistical MRV tool for mCDR. We use OAE as a testbed for the IRF approach, but note this methodology should be suitable for other mCDR intervention strategies, such as direct ocean removal." Similar to the Zhou et al. 2024 study, we are performing simulations of OAE specifically (which is why OAE is in the abstract) but we have noted early in the introduction that the IRF methodology does apply to DOC. The title is already quite long so we have not included "ocean alkalinity enhancement" there.

10-15: this has a stronger statement than warranted, as the approach only tests the fidelity of an IRF for a continuous release within the same model, and the real world should be considered a different "model" entirely. The statement is mostly okay, but should be qualified and moderated.

Noted, we say "in a global ocean model" meaning one model. We have devoted significant discussion to the caveat that this is just one model and that IRFs work well as a proof-of-concept there but acknowledge this model has its own biases/assumptions. Please see the Discussion & Conclusions section. Part of this work is guidance on how to simplify MRV given a set of numerical simulations (how to extend those results to an arbitrary forcing with the assumption that we trust the model), there is more work that needs to be done on model intercomparisons and examining physical influences of resolution and other model assumptions. We discuss this at length in the revisions.

30: the sentence beginning with "A thermostat..." is possibly missing multiple words or is just incorrect.

We believe this sentence is grammatically correct, it is saying that a natural thermostat operates in the climate system. However, we have rephrased this to avoid awkwardness.

39: Jumping around timescales here is problematic, as the premature subduction of TA is irrelevant on the timescales that are relevant for the Earth system feedbacks that were the focus of the beginning of the paragraph. Consider dropping the discussion of the natural thermostat to save length and to focus on how mCDR methods tend to create pCO2 deficits in the surface ocean relative to unmodified conditions, how mCDR doesn't happen until the deficit is eliminated by air-sea exchange, and how subduction slows this equilibration (to timescales that are too slow for climate mitigation strategies).

We appreciate this comment. The motivation for the natural thermostat text is to emphasize that OAE is attempting to accelerate a natural phenomenon; we think this is an important point given the reticence many feel towards geoengineering. Also we don't agree that premature subduction of TA is irrelevant on the timescales that are relevant for Earth system feedbacks. If TA is injected in the Labrador Sea, we don't see it back at the surface for a few thousand years, and the Earth system feedbacks can be quite a bit faster than that.

53: a strategy can consider multiple scales.

The message here is that it's hard to observe a small signal accurately over the entire ocean using a single modeling approach/observations/general technique. We've modified the language from strategy to technique.

57: This implies that computation is our main limitation for modeling. I would argue it is process parameterization and understanding.

Absolutely agreed that process understanding and parameterization is the primary challenge (that will always be a challenge regardless of how high-resolution ocean models become). The sentence reads "the inherent difficulties in **representing the ocean component of the climate and its interactions with the atmosphere, land, and biosphere** using finite computational resources". "Representing the ocean component and its interactions" encapsulates parameterizations and process understanding (necessary for physics-based parameterizations). Having worked a lot on ocean parameterization development, "representation" is often used as a synonym for "parameterization" which is why I chose that language here. If you'd like to see this modified, please let us know what language you'd prefer to see.

58: finish the point by explaining why we don't do counterfactual experiments

Thanks for pointing this out, we rearranged this sentence so it makes more logical sense, it's now placed before saying that we need modeling studies (we can't have counterfactual observations).

65: There is a disconnect in the community currently with some researchers deliberately avoiding the use of the word "efficiency" to refer to nu with others continuing to use the term to refer to nu. The argument against this term is that there is not a 1:1 equivalence between DIC and TA so the DeltaDIC excess relative to DeltaTA does not fit within the "wasted work" paradigm typically reserved for the term (in)efficiency. Several recent publications have instead taken to using efficiency to refer to the fraction of the expected DIC increase from thermodynamic equilibria that has been achieved, e.g., https://iopscience.iop.org/article/10.1088/1748-9326/ad7477/meta, https://essopenarchive.org/doi/full/10.22541/essoar.170957083.34212619, https://agupubs.onlinelibrary.wiley.com/doi/full/10.1029/2022EF002816

It would be helpful if the authors would adopt this practice or remark on why they do not.

We understand your point, we used the conventions presented in Zhou et al. 2024 but have noticed that this convention sometimes leads to confusion. We have added text clarifying our choice below Equation 1.

The first schematic left me confused. Figure 1C, why is the Alk input varying over time if the alk pulse was instantaneous at t=1 in the subplot in A (the answer is implied in the main text, but opaque from the figure and its caption)? What is the color representing in A? Why does t sometimes have a prime?

In subplot A, we probe the system with the impulse, and in panel C we use the IRF to predict the effect of an **arbitrary time series** of alkalinity forcing. That is what is meant by "an arbitrary OAE deployment". We have edited the figure caption to address these points and make it clearer. t' is now also defined in Section 2.

79-88: Why are these conditions important? I can guess, but it should be stated.

These are the mathematical requirements for invoking the impulse response function convolution integral. We rewrote this to: "The application of IRFs to CDR quantification hinges upon two mathematical requirements: linearity and time invariance."

106: it is not clear how this is addressed in Figure 2.

Writing the uptake curve in the form of Equation 4 allows one to bypass performing additional model simulations to obtain the uptake curve. Instead, one can simply perform the convolution pictured in Figure 2. Figure 2 shows that the convolution provides an alternative to perform additional model integrations. We've modified the text to make this clearer.

125-130: don't IRFs typically include some degree of subduction and re-emergence?

Of course; as we say in those lines we first consider purely the "chemistry" problem and then add in ocean dynamics (which of course include subduction and re-emergence). We've modified the beginning of Section 3 to make it clear that we do indeed consider the flow physics (subduction and re-emergence) and biological feedbacks as a modulation to the chemical process of carbon uptake.

Figure 3: how different would this figure and figure 4 be if the atmospheric pCO2 were not fixed?

Please see the recent paper by Tyka (2024):

https://egusphere.copernicus.org/preprints/2024/egusphere-2024-2150/

The effect on the uptake curves is addressed there and does not affect the validity of the results presented here. See the discussion citing Tyka (2024) in Section 6 (and earlier, in Section 3.2, as requested).

175: if we don't expect an impact on pCO2atm then why are we doing mCDR? (partially addressed a few lines later, but this should be addressed immediately)

The assumption isn't that OAE yields no impact on atmospheric pCO2, but rather that for these small deployments we can accurately capture the first-order carbon uptake curve by assuming a non-interactive atmosphere (to make the modeling more affordable). We added an earlier citation to Tyka (2024) and brought this up immediately as suggested (Section 3.2).

215: this argument cuts both ways. Having a small impact means that a small carbon cycle change from a small and local perturbation to the ecosystem function could result in a significant fractional loss in the expected impact.

Agreed; but this part of the text is just describing how we envision modeling OAE deployments and the need for higher-resolution modeling to capture the short-term plume evolution. Is there a specific modification you'd like us to make here? The point we make here is that the "efficiency" is measured **relative** to a direct air removal. The earth system feedback is common to all negative emission technologies and we're intentionally factoring those out, looking only at the relative, intrinsic efficiency of OAE.

256: there are a lot of methods mixed in these results with a fair bit of repetition. It would be best to bring them together and remove them from the results.

Noted, we have streamlined the text in these sections to avoid repetition, but since the IRF methodology is part of the results, we've kept the structure roughly the same.

293: that appears to be a 50% increase in nu if I am reading it correctly?

Sorry for the confusion; the goal was to say it's 1.5x larger (150%), but yes, it's a 50% increase. We have fixed this.

309: On the contrary, to my eye, there appears to be more seasonal variability in 4 realizations than in 14 interannual ensemble members. Please explore this point quantitatively rather than visually. Visually, the point might be more easily seen without the 3 sigma envelope. The two sources of variability appear quantitatively dissimilar at this point in the manuscript... they only appear similar in the context of the later figures that the reader has not yet encountered. It might be better to do a comparison across regions initially.

We've removed that text and agree it's better to save the variability discussion for later where we explicitly quantify the standard deviations.

310: presumably, all variants converge over infinite time, though it is interesting that the seasonal variations seem to converge more slowly

Agreed, though here we make the point that there's convergence on the time scale of several years. Also note that this is not the case for all locations (such as the Labrador Sea) where variance increases through the end of the simulation.

313: this has started to address one of many concerns for the IRF method. This claim is too strong.

The IRF method hinges upon two requirements - linearity and time invariance. We have shown that linearity is not a significant concern, but the time invariance poses a challenge. It is a fair statement to say that if time invariance is accounted for then the IRF method is valid; our findings and figures support that claim. Please elaborate on which part of this is too strong.

324: This seems problematic unless the authors feel that in practice it is likely that IRFs will be computed from the same year as the release. If such simulations are available, then why bother with IRFs at all? Wouldn't it be better to define an ensemble of IRFs for these locations and then test them against an ensemble of releases in various years? The strength of IRFs is that they can be "precomputed" and used later.

That's a great point (one we've also asked ourselves), and we addressed this when we considered the multiyear release experiments as well as the ensembles looking at interannual variability. In the multiyear release experiments, the IRF is ONLY obtained for year 1, that same IRF is used for the entire 5 years of the alkalinity release. In other words, we're using an IRF from a different year than alkalinity is being deployed in for 4 years; though this prediction obviously doesn't perform as well as using the IRF from the same year as the release, it still creates a remarkably accurate prediction as our results show. Looking at Figure 10, as well as Figure 12, we see that the IRF from year one reconstructs the uptake curves to maximum a few percent error for the 5 year releases. We argue that this is due to the standard deviation between ensemble members decreasing with time at most locations. Note that this is not always the case. In the Labrador Sea for instance, the standard deviation increases over the entire time period considered here, and the IRF from year 1 does not do very well in reproducing the 5 year continuous release results. However, this is a fairly anomalous case and we find that in most locations we can account for the variability (time variance) sufficiently just by using a seasonally varying IRF from one year.

We've added an extra paragraph addressing this question in Section 5.4 (second paragraph).

Figure 12... nice figure!

**Thank you!**

345: most of this belongs (and is repeated in) in the caption

Agreed, we've removed that text.

346: This seems to significantly undercut the utility of these results.

Which aspect of this undercuts the utility? We are honest in saying that the aim isn't really to have an IRF for the same year as the release (there's less utility in that), but to use the IRF to predict the evolution of subsequent years. We do this with the 5-year release case. The IRF from year 1 is used to predict the uptake of all the subsequent years, and does so successfully in most regions.

360: generally this section is well written, but the phrasing of this initial statement is too strong

We have added plenty of caveats to this section and also explain that this is a proof of concept that works quite well in this model.

375: or coastal processes, which may be significant for the many proposed coastal mCDR approaches.

Noted, added.

376: lead to or prevent

Noted, added.

**Reviewer 2:**

This manuscript explores the use of impulse response functions (IRFs) to quantify carbon dioxide removal (CDR) through ocean alkalinity enhancement (OAE), a promising marine CDR method. The authors test the IRF approach across various oceanic conditions, finding that it can reliably approximate carbon uptake with minimal error.

Major Concerns:

**1. Simplified Assumptions in IRFs**

The authors used IRFs to predict OAE-CDR efficiency from a chemical perspective, relying on assumptions of linearity and time invariance. These IRF predictions were then compared to the outcomes of a coarse Earth System Model version 2 (ESMV2) simulation. However, this comparison is effectively circular (at least to me): the IRFs do not account for water mass exchanges or mesoscale processes, and the coarse ESM also omits these physical dynamics. As a result, the match between the two datasets is unsurprising, as both primarily reflect chemical hydrodynamics without fully considering the physical processes. It's important to emphasize that without capturing fine-scale physical variability—such as water currents, downwelling, and other dynamic features—the comparison may be less robust than it appears.

There is a misinterpretation of the IRF assumptions in this comment. We do not only treat OAE from a chemical perspective, that was just the first step for deriving the maximum efficiency (Section 3.1) and considering chemistry-induced nonlinearities. Please see the two sections directly after that, Sections 3.2 and 3.3. The flow physics and variability contribute significantly to introducing time variance, which makes the IRF methodology tricky to apply. This is why we create a seasonally varying IRF library and also perform ensembles to consider the interannual variability. We are using the CESM 1 degree model, which does simulate ocean biogeochemistry (through MARBL) and has fidelity in reproducing oceanic variability and current systems. The large scale ocean circulation is captured in this model (albeit mesocale and submesoscale dynamics are not resolved, see our response to your point 3). This model does account for physical dynamics, just not the fine-scale variability. We entirely agree that it is important to extend this work to higher resolution models, and we are in the process of doing so. We used the OAE Atlas of Zhou et al. 2024 as a starting point for our IRF study. As discussed in Zhou et al., there are many physical differences in mixing, seasonality, and subduction/advection of alkalinity present in this model, and we are excited by the success of IRFs here. It will indeed be interesting to examine how the finer-scale physical variability will affect the problem. We want to emphasize that at each location the IRF encapsulates the physical dynamics at that location, i.e. the IRFs themselves deviate from nmax based on the physical subduction/mixing/circulation present at that location. We are comparing IRFs derived from the CESM model for 1998 to continuous releases of alkalinity in that same model extending for several years post-1998 (that's the exciting matchup that we discuss in the later figures). We modify the language in the revisions to avoid confusion on this point and better caveat the lack of mesoscale/submesoscale variability.

We have modified the start of Section 3 to avoid this misinterpretation and emphasize that we do in fact consider the physical dynamics when constructing IRFs.

**2. Figure 12c and the Overlap with nmax**

The close match in Figure 12c is also expected. After five years of TA release, most of the ocean surface has reached or close to equilibrium (based on Figures 9–11) through sea-air exchange, either quickly or slowly. Essentially, this comparison in Figure 12 is like comparing ηmax after considering seasonal and interannual dynamics, which could be determined by simpler methods using datasets like GLODAPv2 (as shown in Figure 3a). Thus, I question the added value of using IRFs to predict CDR efficiency over longer timescales (years) and across much larger spatial coverage (~100 km), when not accounting the physical processes that prevent reaching ηmax.

Please see the prior comment; it's not a matchup with the chemistry-derived ηmax but rather a close matchup of the seasonal IRFs derived for each polygon with the 5-year continuous alkalinity releases that we're showing in Figure 12. The IRFs do encapsulate the physical dynamics specific to that region. Please see Figure 7 for example. The IRF curves are lower than ηmax in cases of seasonal variability and subduction of water masses. Many of the locations we consider do not achieve ηmax and that's not the comparison that we're performing here. The GLODAP data in section 2.1 again is just to compute ηmax and test the nonlinearity of large alkalinity perturbations. We are attempting to encapsulate flow and seasonal variability into our IRF predictions; in some locations where ηmax is achieved rapidly this isn't as important, but in most places we do need to construct an IRF that accounts for flow variability. Since this point was missed we have attempted to make this clearer in the revisions. We also added text in Section 5.4 discussing the question of "how long is an IRF valid for" (addressing the longer timescale point raised here).

**3. Limited Application for Mesoscale Processes**

Given the limitations of IRFs, I also question the claim that IRFs can guide future regional modeling efforts aimed at resolving mesoscale turbulence or submesoscale dynamics (as suggested in lines 400 of the text). Since the IRFs were not designed to capture these more complex physical processes, their applicability to regional models aiming to resolve such variability seems quite constrained.

This is an open question and one that we agree needs to be explored. We are presently performing regional simulations to test this hypothesis and this will be the subject of future work. However, if we consider the OAE Atlas of Zhou et al. 2024, the IRF methodology provides a route by which those results may be extended. The fact that the model doesn't capture mesoscale processes does not invalidate all work stemming from that model. Instead, we should view this as a starting point which can be built upon and refined in future work. Since we constructed the IRFs from that model, indeed they will not predict how mesoscale and submesoscale dynamics will affect uptake. However, we can study that in higher resolution models and apply what we learn to refine how we use IRFs and how we interpret results from the OAE atlas (i.e. does mesoscale/submesoscale turbulence decrease efficiency? does it increase variability? do we need additional ensembles to encapsulate turbulent variability into our IRFs? etc.). We cannot do everything in one study; nonetheless the IRFs are shown to be a promising strategy to be explored further in higher resolution models. We discuss these limitations in the Discussion and Conclusions, and more clearly state that the IRF results here are a proof-of-concept and starting point for extension into other modeling frameworks or model inter-comparisons.

Additionally, Equation (5) appears incorrect. As currently written,  $\eta(t)$  would never approach  $\eta$ max but reach to 0 as time increases, which is problematic for the model.

Thanks for catching this, we've fixed this.

**Reviewer 3:**

**General comments**

The authors developed the use of impulse response functions (IRFs) for predicting the carbon uptake from ocean alkalinity enhancement (OAE) interventions and tested the accuracy of their approach against simulations of continuous-release OAE scenarios using a Community Earth System Model. They found that IRFs sufficiently meet the requirements of linearity and time invariance that they can predict the carbon uptake from continuous alkalinity release scenarios within several percent error and suggest that IRFs may be a viable approach for Monitoring, Reporting and Verification (MRV) of OAE interventions. A key advantage of the IRF approach is that the pre-computed functions can greatly simplify the estimation of carbon uptake from OAE without the need for full biogeochemical simulations. Although the manuscript is generally worthy and well-written, there are some major issues that the authors need to address before publication.

We appreciate the feedback and positive assessment; please see our responses below. We have edited the manuscript to caveat the use of the 1-degree CESM better and added a paragraph to Section 5.4 addressing the question of how long an IRF is valid for.

**Major issues:**

Lack of independent model validation: As noted by another reviewer, the IRFs were developed and tested using the same ESM and thus have many of the same biases in their representation of physical and biogeochemical processes. The resulting agreement between the IRFs and model simulations should therefore be interpreted cautiously. The authors should address this issue and explicitly discuss the physical/biogeochemical processes and feedbacks represented and not represented in the models and potential sources of biases in the IRFs.

Agreed; we have dedicated additional discussion to potential model biases and clearly stating what the model is missing. Please see the Discussion and Conclusions section. We have mentioned that the model does not resolve mesoscale/submesoscale activity and therefore the variability in the model is less than in the real ocean. We also state that CESM uses the biogeochemistry model MARBL (those interested can look into references). We extensively cite the Zhou et al. 2024 study which has now been published and utilizes the NCAR CESM 1 degree model. Our work uses the same model as a starting point for testing the IRF approach in order to extend the OAE Atlas to arbitrary alkalinity release durations and magnitudes. As numerous scientific literature exists that makes use of this model; it is beyond the scope of our work to "validate" this model. However, we entirely agree we should address the potential biases it may introduce to the IRF approach and have elaborated.

To add: one of our goals is to provide a mathematical framework that generalizes the results of a given model to additional scenarios (this is a proof of concept that can work for any model).

**Period of validity of IRFs:**

Can the authors speak on the performance of IRFs older than 5 years and provide recommendations on their reliable application? When should IRFs be recomputed?

This is an interesting question and one that we can only speculate on, given that our simulations run for a maximum of 20 years. In some regions, such as the Labrador Sea, we find that interannual variations are very substantial and the IRF approach leads to relatively high errors even with a 5-year old IRF. Other regions in the ocean exhibit smaller interannual variability on the timescale of years-decades. The frequency with which IRFs need to be recomputed will vary depending on the variability of the geographic region under consideration and the model that's used to compute the IRFs. The relatively laminar 1-degree CESM exhibits less turbulence and flow variability than a higher-resolution (mesoscale or submesoscale permitting) model. Also, we found that the standard deviation in ensemble members generally decreases in most regions (not the case everywhere, such as the Labrador Sea). Based on the CESM results, it appears that using an IRF library from one year and then performing an alkalinity release for 5-10 years afterwards will lead to a reasonable prediction in most regions. One must of course consider the background oceanic and atmospheric variability that ultimately sets the OAE efficiency at a given release location. We are now exploring in greater detail the extent to which IRFs vary across different initial condition scenarios, and model resolutions.

We have added a paragraph at the end of Section 5.4 discussing this point.

**Confusing mathematical notation:**

As noted by other reviews and a public comment, the mathematical notation is confusing and there are some errors. The switch between  $\delta$  and h(t) is confusing, and it is not clear what h(t) is referring to. The manuscript refers to both h(t) and y(t) =  $\eta$ (t) as "impulse response functions."

Equation 5 should be  $\eta(t) = \eta \max - \exp(-t/\tau)$  as in Zhou et al. (2024).

What does the prime notation in t' refer to?

Noted, thank you for catching these errors. We have corrected and clarified the mathematical notation in the revisions. The t' is a dummy integration variable for time. We have differentiated between the variables, see in particular the text above Equation 4.

**Minor comments:**

Calculation of  $\eta$ max (Lines 148-151): Please provide a complete description of how  $\eta$ max is calculated (i.e., that  $\Delta$ DIC is the difference between the initial DIC and the final DIC after the alkalinity perturbation and complete equilibration with atmospheric p(CO2)). What are the assumptions made about atmospheric p(CO2) in the calculations and the values used?

Thanks for the comment, we've expanded upon this as suggested. We also state in the manuscript that atmospheric pCO2 is assumed constant at 425ppm. We've specified that we use a carbonate chemistry calculation package in the Code and Data Availability section, and complete calculations may be found in the Jupyter notebook on Zenodo.

Fig. 4: Is this computed at a constant temperature and salinity or with the in situ values from the GLODAP dataset?

In situ values from GLODAP, we have clarified in the caption.

Fig. 6 and Lines 272-274: What is the reason for the 5 year curve in the lower right panel for  $\eta(t)$  being different from the others, and why aren't there visible kinks on the other curves at the 1 year and 1 month mark?

The data are discretized by 1 month intervals (so there isn't a kink for the 1-month deployments since that's just one data point). If we zoom into the 1-year release curves there are indeed small/negligible kinks at the 1 year mark. They are small because the DIC is increasing rapidly at 1 year, whereas at 5 years there is more of a plateauing in the uptake curve, making the change in delta ALK more obvious. We've added text to clarify the appearance of the kinks.

Line 136: It may be helpful to the reader to include the range of equilibration timescales in the ocean.

We have added this, the text reads: "Typical values for  $\tau$  range from 0.5 to 24 months, with a mean global value of 4.4 months (Jones et al. 2014)". Thanks for the suggestion.

Fig. 7: The color scheme makes it difficult to distinguish the curves for different months.

Distinguishing month by month isn't the main goal of this figure; we just aim to show the general seasonal trends. However, we have modified this to an "hsv" colormap to increase the amount of colors present and improve readability.

**Additional Public Reviewer: Benoît Pasquier**

We have revised Equations 1-3 and the convolution notation accordingly to the publicly posted comments and thank Benoît Pasquier for the very helpful review.

---

## Referee Report (RR1)

Peer review report on "Impulse response functions as a framework for quantifying ocean-based carbon dioxide removal" by Elizabeth Yankovsky, Mengyang Zhou, Michael Tyka, Scott Bachman, David T. Ho, Alicia Karspeck, and Matthew C. Long

**1) General comments:**

I was asked to review the responses to Reviewer 1's concerns; therefore, my main suggestions are based on that request.

From a physical perspective, how can we show that the LTI (Linear Time-Invariant) assumption is a reasonable approximation? Another question I have is: how realistic is it to use alkalinity pulses that last for one month and cover several hundred square kilometers? I wonder how closely this setup reflects real-world applications. Moreover, the biological non-linearity is ignored on the basis of complexity. As a result, the IRF framework is supported only by the LTI assumption from the chemical perspective, which raises concerns about its overall suitability for OAE and mCDR approaches in general.

I believe Reviewer 1's main — that the manuscript "is sufficiently miss-marketed and occasionally overstated that I believe it needs to be rewritten to be shorter, more focused, simpler, and more straightforward in its aims" — has not been fully addressed. While I cannot completely assess Reviewer 1's reasoning behind the recommendation to shorten the manuscript, I agree that presenting this work as a general proof of concept for MRV applications may be too ambitious at this stage. Therefore, I believe additional effort is needed to make the manuscript more digestible and to the point, in line with Reviewer 1's detailed comments.

That said, the authors have, in many cases, provided sound reasoning for the statements questioned and/or made appropriate revisions. I, therefore, recommend the acceptance of the manuscript after the remaining points raised by Reviewer 1 have been more fully addressed.

**2) Specific comments:**

Here, I focus only on those specific comments from Reviewer 1 that I believe were not fully addressed by the authors.

1-5: I understand why the authors chose to focus on OAE; however, I agree with Reviewer 1 that it may be confusing to introduce it so abruptly. The first part of the abstract could be revised to incorporate some of the background information currently presented in the introduction (as the authors mention in their response to this comment). Additionally, including a clarifying sentence at the end of the abstract could help bridge the abstract with the title and reinforce the points raised by the authors in their reply.

10-15: I believe Reviewer 1 was referring to the statements: "We find that the IRF prediction can typically reconstruct the carbon uptake in continuous-release simulations within several percent error. Our simulations elucidate the influences of oceanic variability and deployment duration on carbon uptake efficiency." While this is addressed in the manuscript, I agree that the abstract should make it clear that these findings only refer specifically to the model used by the authors.

39: I agree with Reviewer 1 that the mixing of timescales is problematic. If the main point is that OAE aims to accelerate rock weathering to ultimately enhance CO₂ uptake, then I concur with the reviewer that if the added alkalinity sinks, that objective is not achieved. I understand the authors' intention with

the "thermostat" analogy, but I also agree with the reviewer that this could be omitted to maintain focus on the primary goal of OAE—enhancing CO2 uptake at the ocean surface.

106: The authors state that they have modified the text to clarify their point, but they have only added "h" to that sentence. They could strengthen the manuscript by expanding on this point in more detail—similar to how they do in their response to Reviewer 1.

Figures 3 and 4: The authors have added new text to the discussion section in response to the comment about how different Figures 3 and 4 would be if atmospheric pCO2 were not fixed. However, this addition only partially addresses Reviewer 1's question. There is no clear answer provided regarding how different the figures would actually be. Instead, the authors mention that including an interactive atmosphere and terrestrial carbon pools "may be important" future considerations, as this could reduce the sensitivity of the biological pump to changes in carbon uptake. I believe that simply acknowledging this known limitation, without offering any estimate or indication of how the final results might change, does not fully address the reviewer's concern. One possible solution would be to add a note directly to Figures 3 and 4, flagging this limitation for readers who may focus on the figures without reading the full discussion. However, I still think the authors should include some numbers to their estimation of the error introduced by not having interactive atmosphere and terrestrial carbon pools.

Figures 6, 7, and 8 do not include subplot labels, although they are referenced in the text using subplot letters.

---

## Author Response (AR2)

**Reviewer 1:**

Thank you for providing additional details about the impulse response function (IRF) framework, particularly the role of  $\eta_{\text{max}}$ , which you note is derived based on baseline carbonate chemistry (i.e., the DIC/TA ratio), and how other parameters such as downwelling constrain the system from reaching  $\eta_{\text{max}}$ .

I have a few clarification questions and suggestions:

1. Could the authors clarify how  $\eta_{\text{max}}$  was determined? I assume it was calculated using GLODAP data. If so, does the seasonally varying IRF (and the interannual ensemble predictions) also reflect variability in  $\eta_{\text{max}}$  across seasons or years?

In the introduction, we have highlighted in red the description of how  $\eta_{\text{max}}$  was determined. Indeed, it is done using the GLODAP climatology and does not take into account seasonal variations. Please see the Figure 3 caption and the two paragraphs below equation 6. This is for the purpose of assessing linearity and time invariance.

However, the  $\eta_{\text{max}}$  is not computed a priori when deriving the IRFs. We added text beneath Equation 8 (the IRF model) to emphasize that this is done as a curve-fitting.  $\eta_{\text{max}}$  will indeed vary seasonally and will depend on where the plume goes, thus we do not impose it as a constraint but rather simply fit an analytical IRF curve to the observed curve. Please see the next comment for more explanation. How to conceptually understand " $\eta_{\text{max}}$ " is indeed tricky as the reviewer points out due to the fact that the plume sees many carbonate chemistry states as it propagates out of its initial region and the effective  $\eta_{\text{max}}$  will be a function of time and space.

- 2. Some key parameters in the IRF formulation are currently under-described. For example, in Equation 8, could the authors provide the actual values of  $\tau_1$ ,  $\tau_2$ , and  $\tau_3$ ? Also, do these time constants vary by subregion, or are the same values applied uniformly across the domain? We've expanded upon this, please see the text surrounding Equation 8. There is also clarification about  $\eta_{max}$  there.
- 3. If my understand it right, one of the major advantages of the IRF approach appears to be its ability to quantify whether—and when—OAE might approach  $\eta_{\text{max}}$ . To enhance reader understanding, would the authors consider including  $\eta_{\text{max}}$  as a reference line or visual marker in one or more of the figures? This could help clarify the framework's predictive purpose and the degree to which local or seasonal factors constrain OAE efficiency relative to the theoretical maximum.

We appreciate this comment, but believe the map of  $\eta_{\text{max}}$  shown in Figure 3 accomplishes this goal to first order. Our figures already have a great deal of information with ensembles and seasonal curves, and we prefer to keep the figures as is. Also, it's challenging to identify a single purely chemical  $\eta_{\text{max}}$  for a given alkalinity plume as it propagates through various regions around the globe (experiencing differing ALK, DIC, T, and S values), so we believe Figure 3 will suffice here. More information on  $\eta_{\text{max}}$  and equilibration timescales are presented in Zhou et al. 2024.

**Reviewer 2:**

**1) General comments:**

I was asked to review the responses to Reviewer 1's concerns; therefore, my main suggestions are based on that request.

From a physical perspective, how can we show that the LTI (Linear Time-Invariant) assumption is a reasonable approximation? Another question I have is: how realistic is it to use alkalinity pulses that last for one month and cover several hundred square kilometers? I wonder how closely this setup reflects real-world applications. Moreover, the biological non-linearity is ignored on the basis of complexity. As a result, the IRF framework is supported only by the LTI assumption from the chemical perspective, which raises concerns about its overall suitability for OAE and mCDR approaches in general.

I believe Reviewer 1's main — that the manuscript "is sufficiently miss-marketed and occasionally overstated that I believe it needs to be rewritten to be shorter, more focused, simpler, and more straightforward in its aims" — has not been fully addressed. While I cannot completely assess Reviewer 1's reasoning behind the recommendation to shorten the manuscript, I agree that presenting this work as a general proof of concept for MRV applications may be too ambitious at this stage. Therefore, I believe additional effort is needed to make the manuscript more digestible and to the point, in line with Reviewer 1's detailed comments.

That said, the authors have, in many cases, provided sound reasoning for the statements questioned and/or made appropriate revisions. I, therefore, recommend the acceptance of the manuscript after the remaining points raised by Reviewer 1 have been more fully addressed.

We appreciate the overall positive assessment of our revisions, and address the comments listed below point-by-point.

**2) Specific comments:**

Here, I focus only on those specific comments from Reviewer 1 that I believe were not fully addressed by the authors.

1-5: I understand why the authors chose to focus on OAE; however, I agree with Reviewer 1 that it may be confusing to introduce it so abruptly. The first part of the abstract could be revised to incorporate some of the background information currently presented in the introduction (as the authors mention in their response to this comment). Additionally, including a clarifying sentence at the end of the abstract could help bridge the abstract with the title and reinforce the points raised by the authors in their reply.

We have edited the second sentence to introduce OAE more gently, and the ending to say that the IRF approach is broadly applicable to ocean-based CDR (we discuss the similarity to direct ocean removal later in the manuscript).

10-15: I believe Reviewer 1 was referring to the statements: "We find that the IRF prediction can typically reconstruct the carbon uptake in continuous-release simulations within several percent error. Our simulations elucidate the influences of oceanic variability and deployment duration on carbon uptake efficiency." While this is addressed in the manuscript, I agree that the abstract should make it clear that these findings only refer specifically to the model used by the authors.

Agreed, we've modified the text to include "in our model": "We find that the IRF prediction can typically reconstruct the carbon uptake in continuous-release simulations in our model within several percent error."

39: I agree with Reviewer 1 that the mixing of timescales is problematic. If the main point is that OAE aims to accelerate rock weathering to ultimately enhance CO2 uptake, then I concur with the reviewer that if the added alkalinity sinks, that objective is not achieved. I understand the authors' intention with the "thermostat" analogy, but I also agree with the reviewer that this could be omitted to maintain focus on the primary goal of OAE—enhancing CO2 uptake at the ocean surface.

We appreciate the reviewer's perspective, but really prefer to keep the thermostat analogy. The paragraph includes justification about alkalinity remaining at the surface. If the reviewers feel strongly about rewriting this paragraph we will, but we believe it adds to the narrative to present OAE as the acceleration of the natural silicate weathering cycle.

106: The authors state that they have modified the text to clarify their point, but they have only added "h" to that sentence. They could strengthen the manuscript by expanding on this point in more detail—similar to how they do in their response to Reviewer 1.

We explain this in the end of the Figure 2 caption, by saying "However, provided we have a sufficiently LTI system, we can compute the convolution of the IRF and the forcing, thus avoiding the need for an additional model integration." This is also explained further in lines 108-109 (i.e. how Figure 2 illustrates the strength of the IRF, which is what Reviewer 1 initially asked about).

Figures 3 and 4: The authors have added new text to the discussion section in response to the comment about how different Figures 3 and 4 would be if atmospheric pCO2 were not fixed. However, this addition only partially addresses Reviewer 1's question. There is no clear answer provided regarding how different the figures would actually be. Instead, the authors mention that including an interactive atmosphere and terrestrial carbon pools "may be important" future considerations, as this could reduce the sensitivity of the biological pump to changes in carbon uptake. I believe that simply acknowledging this known limitation, without offering any estimate or indication of how the final results might change, does not fully address the reviewer's concern. One possible solution would be to add a note directly to Figures 3 and 4, flagging this limitation for readers who may focus on the figures without reading the full discussion. However, I still think the authors should include some numbers to their estimation of the error introduced by not having interactive atmosphere and terrestrial carbon pools.

Figures 3 and 4 view OAE as an idealized chemical process, neglecting the nonlinearities of biosphere, atmosphere, and lithosphere interactions. This is acknowledged, and unfortunately we can only speculate what the net effect of including and fully resolving all of these constituents would be (including numbers would be speculative/ unfounded).

However, we have added text to the Figure 3 caption as suggested, saying: "Note: these calculations assume an idealized, non-interactive atmosphere." We have cited work that addresses the influence of an interactive atmosphere and have included caveats several times in the manuscript. For instance, please see lines 185-190: "Although OAE will decrease the

atmospheric pCO2 and thus impact CO2 uptake, for small OAE deployments we can accurately capture the first-order carbon uptake curve by assuming a non-interactive atmosphere, making modeling more affordable (Tyka, 2024)." The Tyka paper does quantify the role of an interactive atmosphere, generally found to be small but the reader may consult that paper for additional information. For the sake of keeping the paper concise (as Reviewer 1 suggested) we've simply cited this paper and briefly discussed the limitations of our non-interactive atmosphere.

Figures 6, 7, and 8 do not include subplot labels, although they are referenced in the text using subplot letters.

Thank you for pointing this out. We identified one instance where this mistake was made (referencing Fig. 7b), and this has been corrected.